# Compressive Visual Representations

**Kuang-Huei Lee**[†]
Google Research
leekh@google.com

**Anurag Arnab**[†]
Google Research
aarnab@google.com

**Sergio Guadarrama**
Google Research
sguada@google.com

**John Canny**
Google Research
canny@google.com

**Ian Fischer**[†]
Google Research
iansf@google.com

## Abstract

Learning effective visual representations that generalize well without human su-pervision is a fundamental problem in order to apply Machine Learning to a wide variety of tasks. Recently, two families of self-supervised methods, contrastive learning and latent bootstrapping, exemplified by SimCLR and BYOL respec-tively, have made significant progress. In this work, we hypothesize that adding explicit information compression to these algorithms yields better and more robust representations. We verify this by developing SimCLR and BYOL formulations compatible with the Conditional Entropy Bottleneck (CEB) objective, allowing us to both measure and control the amount of compression in the learned representa-tion, and observe their impact on downstream tasks. Furthermore, we explore the relationship between Lipschitz continuity and compression, showing a tractable lower bound on the Lipschitz constant of the encoders we learn. As Lipschitz continuity is closely related to robustness, this provides a new explanation for why compressed models are more robust. Our experiments confirm that adding compression to SimCLR and BYOL significantly improves linear evaluation accu-racies and model robustness across a wide range of domain shifts. In particular, the compressed version of BYOL achieves 76.0% Top-1 linear evaluation accuracy on ImageNet with ResNet-50, and 78.8% with ResNet-50 2x.[1]

## 1 Introduction

Individuals develop mental representations of the surrounding world that generalize over different views of a *shared context*. For instance, a shared context could be the identity of an object, as it does not change when viewed from different perspectives or lighting conditions. This ability to represent views by distilling information about the *shared context* has motivated a rich body of self-supervised learning work [54, 4, 12, 30, 33, 47]. For a concrete example, we could consider an image from the ImageNet training set [60] as a shared context, and generate different views by repeatedly applying different data augmentations. Finding stable representations of a shared context corresponds to learning a minimal high-level description since not all information is relevant or persistent. This explicit requirement of learning a concise representation leads us to prefer objectives that are *compressive* and only retain the relevant information.

Recent contrastive approaches to self-supervised visual representation learning aim to learn rep-resentations that maximally capture the mutual information between two transformed views of an image [54, 4, 12, 33, 40]. The primary idea of these approaches is that this mutual information

---

[†]Main contributors

[1]Code available at `https://github.com/google-research/compressive-visual-representations`

35th Conference on Neural Information Processing Systems (NeurIPS 2021).

corresponds to a general shared context that is invariant to various transformations of the input, and it is assumed that such invariant features will be effective for various downstream higher-level tasks. However, although existing contrastive approaches maximize mutual information between augmented views of the same input, they do not necessarily compress away the irrelevant information from these views [12, 33]. As shown in [26, 27], retaining irrelevant information often leads to less stable representations and to failures in robustness and generalization, hampering the efficacy of the learned representations. An alternative state-of-the-art self-supervised learning approach is BYOL [30], which uses a slow-moving average network to learn consistent, view-invariant representations of the inputs. However, it also does not explicitly capture relevant compression in its objective.

In this work, we modify SimCLR [12], a state-of-the-art contrastive representation method, by adding information compression using the Conditional Entropy Bottleneck (CEB) [27]. Similarly, we show how BYOL [30] representations can also be compressed using CEB. By using CEB we are able to measure and control the amount of information compression in the learned representation [26], and observe its impact on downstream tasks. We empirically demonstrate that our compressive variants of SimCLR and BYOL, which we name C-SimCLR and C-BYOL, significantly improve accuracy and robustness to domain shifts across a number of scenarios. Our primary contributions are:

- Reformulations of SimCLR and BYOL such that they are compatible with information-theoretic compression using the Conditional Entropy Bottleneck [26].

- An exploration of the relationship between Lipschitz continuity, SimCLR, and CEB compression, as well as a simple, tractable lower bound on the Lipschitz constant. This provides an alternative explanation, in addition to the information-theoretic view [26, 27, 1, 2], for why CEB compression improves SimCLR model robustness.

- Extensive experiments supporting our hypothesis that adding compression to the state-of-the-art self-supervised representation methods like SimCLR and BYOL can significantly improve their performance and robustness to domain shifts across multiple datasets. In particular, linear evaluation accuracies of C-BYOL are even competitive with the supervised baselines considered by SimCLR [12] and BYOL [30]. C-BYOL reaches 76.0% and 78.8% with ResNet-50 and ResNet-50 2x respectively, whereas the corresponding supervised baselines are 76.5% and 77.8% respectively.

## 2 Methods

In this section, we describe the components that allow us to make distributional, compressible versions of SimCLR and BYOL. This involves switching to the Conditional Entropy Bottleneck (CEB) objective, noting that the von Mises-Fisher distribution is the exponential family distribution that corresponds to the cosine similarity loss function used by SimCLR and BYOL, and carefully identifying the random variables and the variational distributions needed for CEB in SimCLR and BYOL. We also note that SimCLR and CEB together encourage learning models with a smaller Lipschitz constant, although they do not explicitly enforce that the Lipschitz constant be small.

### 2.1 The Conditional Entropy Bottleneck

In order to test our hypothesis that compression can improve visual representation quality, we need to be able to measure and control the amount of compression in our visual representations. To achieve this, we use the Conditional Entropy Bottleneck (CEB) [26], an objective function in the Information Bottleneck (IB) [66] family.

Given an observation $X$, a target $Y$, and a learned representation $Z$ of $X$, CEB can be written as:

$$CEB \equiv \min_{Z} \beta I(X; Z|Y) - I(Y; Z) \tag{1}$$

$$= \min_{Z} \beta(H(Z) - H(Z|X) - H(Z) + H(Z|Y)) - H(Y) + H(Y|Z) \tag{2}$$

$$= \min_{Z} \beta(-H(Z|X) + H(Z|Y)) + H(Y|Z) \tag{3}$$

where $H(\cdot)$ and $H(\cdot|\cdot)$ denote entropy and conditional entropy respectively. We can drop the $H(Y)$ term because it is constant with respect to $Z$. $I(Y; Z)$ is the useful information relevant to the task, or the prediction target $Y$. $I(X; Z|Y)$ is the *residual information* $Z$ captures about $X$ when we already know $Y$, which we aim to minimize. Compression strength increases as $\beta$ increases.

We define $e(z|x)$ as the true encoder distribution, where $z$ is sampled from; $b(z|y)$, a variational approximation conditioned on $y$; $d(y|z)$, the decoder distribution (also a variational approximation) which predicts $y$ conditioned on $z$. As shown in [26], CEB can be variationally upper-bounded:

$$vCEB \equiv \min_{e(z|x),b(z|y),d(y|z)} \mathbb{E}_{x,y\sim p(x,y),z\sim e(z|x)} \beta(\log e(z|x) - \log b(z|y)) - \log d(y|z) \quad (4)$$

There is no requirement that all three distributions have learned parameters. At one limit, a model's parameters can be restricted to any one of the three distributions; at the other limit, all three distributions could have learned parameters. If $e(\cdot)$ has learned parameters, its distributional form may be restricted, as we must be able to take gradients through the $z$ samples.[2] The only requirement on the $b(\cdot)$ and $d(\cdot)$ distributions is that we be able to take gradients through their log probability functions.

**InfoNCE.**    As shown in [26], besides parameterizing $d(y|z)$, it is possible to reuse $b(z|y)$ to make a variational bound on the $H(Y|Z)$ term. As $I(Y;Z) = H(Y) - H(Y|Z)$ and $H(Y)$ is a constant with respect to $Z$:

$$H(Y|Z) \leq E_{x,y\sim p(x,y),z\sim e(z|x)} \log \frac{b(z|y)}{\sum_{k=1}^{K} b(z|y_k)} \quad (5)$$

where $K$ is the number of examples in a minibatch. Eq. (5) is also known as the contrastive *InfoNCE* bound [54, 56]. The inner term,

$$d(y|z) \equiv \frac{b(z|y)}{\sum_{k=1}^{K} b(z|y_k)}, \quad (6)$$

is a valid variational approximation of the true but unknown $p(y|z)$. Fischer [26] calls Eq. (6) the *CatGen* decoder because it is a categorical distribution over the minibatch that approximates the generative decoder distribution.

## 2.2   C-SimCLR: Compressed SimCLR

The InfoNCE bound [54] enables many contrastive visual representation methods to use it to capture shared context between different views of an image as a self-supervised objective [12, 13, 33, 15, 40]. In this work, we show how to compress the SimCLR [12] model, but the method we discuss is generally applicable to other InfoNCE-based models.

SimCLR applies randomized augmentations to an image to create two different views, $x$ and $y$ (which we also refer to as $x'$), and encodes both of them with a shared encoder, producing representations $r_x$ and $r_y$. Both $r_x$ and $r_y$ are $l_2$-normalized. The SimCLR version of the InfoNCE objective is:

$$L_{NCE}(r_x, r_y) = -\log \frac{e^{\frac{1}{\tau} r_y^T r_x}}{\sum_{k=1}^{K} e^{\frac{1}{\tau} r_{y_k}^T r_x}} \quad (7)$$

where $\tau$ is a temperature term and $K$ is the number of views in a minibatch. SimCLR further makes its InfoNCE objective *bidirectional*, such that the final objective becomes

$$L_{NCE}(r_x, r_y) + L_{NCE}(r_y, r_x) = -\log \frac{e^{\frac{1}{\tau} r_y^T r_x}}{\sum_{k=1}^{K} e^{\frac{1}{\tau} r_{y_k}^T r_x}} - \log \frac{e^{\frac{1}{\tau} r_x^T r_y}}{\sum_{k=1}^{K} e^{\frac{1}{\tau} r_{x_k}^T r_y}} \quad (8)$$

We can observe the following: $\exp(\frac{1}{\tau} r_y^T r_x)$ in Eq. (7) corresponds to the unnormalized $b(z|y)$ in Eq. (5). $e(\cdot|x)$ generates $z = r_x$, whilst $r_y$ and $r_{y_k}$ are distribution parameters of $b(\cdot|y)$ and $b(\cdot|y_k)$ respectively. $e(\cdot|x)$ and $b(\cdot|y)$ share model parameters.

**von Mises-Fisher Distributional Representations.**    The cosine-similarity-based loss (Eq. (7)) is commonly used in contrastive learning and can be connected to choosing the von Mises-Fisher (vMF) distribution for $e(\cdot|x)$ and $b(\cdot|y)$ [32, 70]. vMF is a distribution on the $(n-1)$-dimensional hyper-sphere. The probability density function is given by $f_n(z, \mu, \kappa) = C_n(\kappa)e^{\kappa \mu^T z}$, where $\mu$ and $\kappa$ are denoted as the mean direction and concentration parameter respectively. We assume $\kappa$ is a

---

[2]For example, $e(z|x)$ could not generally be a mixture distribution, as sampling the mixture distribution has a discrete component, and we cannot easily take gradients through discrete samples.

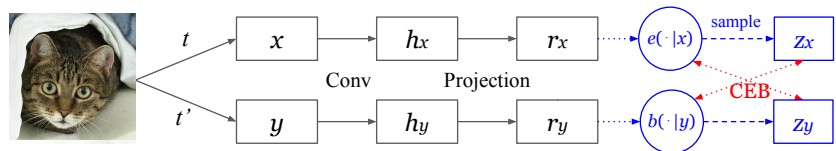

Figure 1: C-SimCLR explicitly defines encoder distributions $e(\cdot|x)$ and $b(\cdot|y)$ where $x$ and $y$ are two augmented views of an image. $y$ is also referred as $x'$. The upper and lower encoder outputs are used to specify mean directions of $e$ and $b$, and the two encoders share parameters. $r_x, r_y$ are $l_2$-normalized. Our modifications to SimCLR are highlighted in blue. No new parameters are added.

constant. The normalization term $C_n(\kappa)$ is a function of $\kappa$ and equal to $\frac{\kappa^{n/2-1}}{(2\pi)^{n/2}I_{n/2-1}(\kappa)}$, where $I_v$ denotes the modified Bessel function of the first kind at order $v$.

By setting the mean direction $\mu$ to $r_y$, concentration $\kappa_b$ of $b(\cdot|y)$ to $1/\tau$, and $r_x$ to $z$, we can connect the SimCLR objective (Eq. (7)) to the distributional form of InfoNCE (Eq. (5))

$$\frac{e^{\frac{1}{\tau}r_y^T r_x}}{\sum_{k=1}^{K} e^{\frac{1}{\tau}r_{y_k}^T r_x}} = \frac{C_n(\kappa_b)e^{\kappa_b r_y^T r_x}}{\sum_{k=1}^{K} C_n(\kappa_b)e^{\kappa_b r_{y_k}^T r_x}} = \frac{f_n(r_x, r_y, \kappa_b)}{\sum_{k=1}^{K} f_n(r_x, r_{y_k}, \kappa_b)} = \frac{b(r_x|y)}{\sum_{k=1}^{K} b(r_x|y_k)} \quad (9)$$

$z = r_x$ is a deterministic unit-length vector, so we can view $e(\cdot|x)$ as a spherical delta distribution, which is equivalent to a vMF with $r_x$ as the mean direction and $\kappa_e \to \infty$. We can further extend the forward encoder to have non-infinite $\kappa_e$, which results in a stochastic $z$. These allow us to have SimCLR in a distributional form with explicit distributions $e(\cdot|x)$ and $b(\cdot|y)$ and satisfy the requirements of CEB discussed in Sec. 2.1.

**Compressing SimCLR with Bidirectional CEB.** Figure 1 illustrates the Compressed SimCLR (C-SimCLR) model. The model learns a compressed representation of an view $X$ that only preserves information relevant to predicting a different view $Y$ by switching to CEB. As can be seen in Eq. (3), the CEB objective treats $X$ and $Y$ asymmetrically. However, as shown in [26], it is possible to learn a single representation $Z$ of both $X$ and $Y$ by having the forward and backward encoders act as variational approximations of each other:

$$CEB_{\text{bidir}} \equiv \min_Z \beta_X I(X; Z|Y) - I(Y; Z) + \beta_Y I(Y; Z|X) - I(X; Z) \quad (10)$$

$$\equiv \min_Z \beta_X(-H(Z|X) + H(Z|Y)) + H(Y|Z) \quad (11)$$

$$+ \beta_Y(-H(Z|Y) + H(Z|X)) + H(X|Z)$$

$$\leq \min_{e(\cdot|\cdot), b(\cdot|\cdot), c(\cdot|\cdot), d(\cdot|\cdot)} \mathbb{E}_{x,y \sim p(x,y)} \Big[ \quad (12)$$

$$\mathbb{E}_{z_x \sim e(z_x|x)} \big[ \beta_X (\log e(z_x|x) - \log b(z_x|y)) - \log d(y|z_x) \big]$$

$$+ \mathbb{E}_{z_y \sim e(z_y|y)} \big[ \beta_Y (\log(e(z_y|y) - \log(b(z_y|x)) - \log c(x|z_y)) \big] \Big]$$

where $d(\cdot|\cdot)$ and $c(\cdot|\cdot)$ are the InfoNCE variational distributions of $b(\cdot|\cdot)$ and $e(\cdot|\cdot)$ respectively. $e$ and $b$ use the same encoder to parameterize mean direction in SimCLR setting. Since SimCLR is trained with a bidirectional InfoNCE objective, Eq. (12) gives an easy way to compress its learned representation. As in SimCLR, the deterministic $h_x$ (in Fig. 1) is still the representation used on downstream classification tasks.

### 2.3 C-BYOL: Compressed BYOL

In this section we will describe how to modify BYOL to make it compatible with CEB, as summarized in Fig. 2. BYOL [30] learns an online encoder that takes $x$, an augmented view of a given image, as input and predicts outputs of a target encoder which encodes $x'$, a different augmented view of the same image. The target encoder's parameters are updated not by gradients but as an exponential moving average of the online encoder's parameters. The loss function is simply the mean square error, which is equivalent to the cosine similarity between the online encoder output $\mu_e$ and the target encoder output $y'$ as both $\mu_e$ and $y'$ are $l_2$-normalized:

$$L_{byol} = ||\mu_e - y'||_2^2 = \mu_e^T \mu_e + y'^T y' - 2\mu_e^T y' = 2 - 2\mu_e^T y' \quad (13)$$

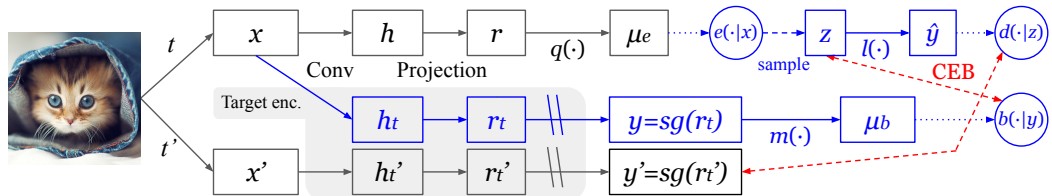

Figure 2: C-BYOL. The upper online encoder path takes an augmented view $x$ as input and produces $e(\cdot|x)$ and $d(\cdot|z)$. The lower two paths use the same target encoder (shaded), which is a moving average of the online encoder (Conv + Projection). The target encoder maps $x$ and another view $x'$ to $r_t$ and $r'_t$. $sg(r_t)$ ($sg$: stop gradients) is our target $y$. $y$ leads to $b(\cdot|y)$. $sg(r'_t)$ is our perturbed target $y'$. $r_t, r'_t, \mu_e, \mu_b, \hat{y}$ are $l_2$-normalized. These yield the components required by CEB. We highlight changes to BYOL in blue.

This iterative "latent bootstrapping" allows BYOL to learn a view-invariant representation. In contrast to SimCLR, BYOL does not rely on other samples in a batch and does not optimize the InfoNCE bound. It is a simple regression task: given input $x$, predict $y'$. To make BYOL CEB-compatible, we need to identify the random variables $X, Y, Z$, define encoder distributions $e(z|x)$ and $b(z|y)$, and define the decoder distribution $d(y|z)$ (see Equation (4)).

We define $e(z|x)$ to be a vMF distribution parameterized by $\mu_e$, and sample $z$ from $e(z|x)$:

$$e(z|x) = C_n(\kappa_e)e^{\kappa_e z^T \mu_e} \tag{14}$$

We use the target encoder to encode $x$ and output $r_t$, an $l_2$-normalized vector. We choose $r_t$ to be $y$. We then add a 2-layer MLP on top of $y$ and $l_2$-normalize the output, which gives $\mu_b$. We denote this transformation as $\mu_b = m(y)$ and define $b(z|y)$ to be the following vMF parameterized by $\mu_b$:

$$b(z|y) = C_n(\kappa_b)e^{\kappa_b z^T \mu_b} \tag{15}$$

For $d(y|z)$, we add a linear transformation on $z$ with $l_2$-normalization, $\hat{y} = l(z)$, and define a vMF parameterized by $\hat{y}$:

$$d(y|z) = C_n(\kappa_d)e^{\kappa_d y^T \hat{y}} \tag{16}$$

In the deterministic case where $z$ is not sampled, this corresponds to adding a linear layer with $l_2$-normalization on $\mu_e$ which does not change the model capacity and empirical performance.

In principle, we can use any stochastic function of $Z$ to generate $Y$. In our implementation, we replace the generative decoder $\log d(y|z)$ with $\log d(y'|z)$, where we use the target encoder to encode $x'$ and output $y'$. Given that $X \to X'$ is a stochastic transformation and both $X$ and $X'$ go through the same the target encoder function, $Y \to Y'$ is also a stochastic transformation. $d(y'|z)$ can be considered as having a stochastic perturbation to $d(y|z)$. Our $vCEB$ objective becomes

$$L_{cbyol}(x, x') = \beta(\log e(z|x) - \log b(z|y)) - \log d(y'|z). \tag{17}$$

We empirically observed the best results with this design choice. $d(y'|z)$ can be directly connected the standard BYOL regression objective: When $\kappa_d = 2$, $-\log(d(y'|z)) = -\kappa_d y'^T \hat{y} - \log(C_n(\kappa_d))$ is equivalent to Eq. (13) when constants are ignored.

Although it seems that we additionally apply the target encoder to $x$ compared to BYOL, this does not increase the computational cost in practice. As in BYOL, the learning objective is applied symmetrically in our implementation: $L_{cbyol}(x, x') + L_{cbyol}(x', x)$. Therefore, the target encoder has to be applied to both $x$ and $x'$ no matter in BYOL or C-BYOL. Finally, note that like in BYOL, $h$ (Fig. 2) is the deterministic representation used for downstream tasks.

## 2.4 Lipschitz Continuity and Compression

Lipschitz continuity provides a way of measuring how smooth a function is. For some function $f$ and a distance measure $D(f(x_1), f(x_2))$, Lipschitz continuity defines an upper bound on how quickly $f$ can change as $x$ changes:

$$L||\Delta x|| \geq D(f(x), f(x + \Delta x)), \tag{18}$$

where $L$ is the Lipschitz constant, $\Delta x$ is the vector change in $x$, and $||\Delta x|| > 0$. If we define $f(x)$ to be our encoder distribution $e(z|x)$ (which is a vMF and always positive), and the distance measure, $D$, to be the absolute difference of the logs of the functions, we get a function of $z$ of Lipschitz value, such that:

$$L(z) \geq \frac{1}{||\Delta x||}|\log e(z|x) - \log e(z|x + \Delta x)| \tag{19}$$

As detailed in Sec. G, by taking expectations with respect to $z$, we can obtain a lower bound on the encoder *distribution*'s squared Lipschitz constant:[3]

$$L^2 \geq \frac{1}{||\Delta x||^2} \max\Big( \mathrm{KL}[e(z|x)||e(z|x + \Delta x)], \, \mathrm{KL}[e(z|x + \Delta x)||e(z|x)]\Big) \tag{20}$$

To guarantee smoothness of the encoder distribution, we would like to have an upper bound on $L$, rather than a lower bound. Minimizing a lower bound does not directly yield any optimality guarantees relative to the bounded quantity. However, in this case, minimizing the symmetric KL below is *consistent* with learning a smoother encoder function:

$$\inf_{e(z|\cdot)} \mathrm{KL}[e(z|x)||e(z|x + \Delta x)] + \mathrm{KL}[e(z|x + \Delta x)||e(z|x)] \tag{21}$$

By *consistent*, we mean that, if we could minimize this symmetric KL at every pair $(x, x + \Delta x)$ in the input domain, we would have smoothed the model. In practice, for high-dimensional input domains, that is not possible, but minimizing Eq. (21) at a subset of the input domain still improves the model's smoothness, at least at that subset.

The minimization in Eq. (21) corresponds almost exactly to the CEB compression term in the bidirectional SimCLR models. We define $y = x + \Delta x$. At samples of the augmented observed variables, $X, Y$, the C-SimCLR models minimize upper bounds on the two residual informations:

$$I(X; Z|Y) + I(Y; Z|X) \leq \mathbb{E}_{x,y \sim p(x,y)} \mathrm{KL}[e(z|x)||e(z|y)] + \mathrm{KL}[e(z|y)||e(z|x)] \tag{22}$$

The only caveat to this is that we use $b(z|y)$ instead of $e(z|y)$ in C-SimCLR. $b$ and $e$ share weights but have different $\kappa$ values in their vMF distributions. However, these are hyperparameters, so they are not part of the trained model parameters. They simply change the minimum attainable KLs in Eq. (22), thereby adjusting the minimum achievable Lipschitz constant for the models (see Sec. G).

Directly minimizing Equation (20) would require normalizing the symmetric KL per-example by $||\Delta x||^2$. The symmetric CEB loss does not do this. However, the residual information terms in Equation (22) are multiplied by a hyperparameter $\beta \leq 1$. Under a simplifying assumption that the $||\Delta x||$ values generated by the sampling procedure are typically of similar magnitude, we can extract the average $\frac{1}{||\Delta x||^2}$ into the hyperparameter $\beta$. We note that in practice using per-example values of $||\Delta x||^2$ would encourage the optimization process to smooth the model more strongly at observed $(x, \Delta x)$ pairs where it is least smooth, but we leave such experiments to future work.

Due to Eq. (22), we should expect that the C-SimCLR models are locally more smooth around the observed data points. We reiterate, though, that this is not a proof of increased global Lipschitz smoothness, as we are minimizing a lower bound on the Lipschitz constant, rather than minimizing an upper bound. It is still theoretically possible to learn highly non-smooth functions using CEB in this manner. It would be surprising, however, if the C-SimCLR were somehow *less* smooth than the corresponding SimCLR models.

The Lipschitz continuity property is closely related to model robustness to perturbations [9], including robustness to adversarial examples [71, 23, 73]. Therefore, we would expect to see that the C-SimCLR models are more robust than SimCLR models on common robustness benchmarks. It is more difficult to make the same theoretical argument for the C-BYOL models, as they do not use exactly the same encoder for both $x$ and $y$. Thus, the equivalent conditional information terms from Eq. (22) are not directly minimizing a lower bound on the Lipschitz constant of the encoder. Nevertheless, we empirically explore the impact of CEB on both SimCLR and BYOL models next in Sec. 3.

---

[3]Note that by taking an expectation we get a KL divergence, which violates the triangle inequality, even though we started from a valid distance metric. Squaring the Lipschitz constant addresses this in the common case where the KL divergence grows quadratically in $||\Delta x||$, as detailed in Section G.

# 3 Experimental Evaluation

We first describe our experimental set-up in Sec. 3.1, before evaluating the image representations learned by our self-supervised models in linear evaluation settings in Sec. 3.2. We then analyse the robustness and generalization of our self-supervised representations by evaluating model accuracy across a wide range of domain and distributional shifts in Sec. 3.3. Finally, we analyse the effect of compression strength in Sec. 3.4. Additional experiments and ablations can be found in the Appendix.

## 3.1 Experimental Set-up

**Implementation details.** Our implementation of SimCLR, BYOL, and their compressed versions is based off of the public implementation of SimCLR [12]. Our implementation consistently reproduces BYOL results from [30] and outperforms the original SimCLR, as detailed in Sec. A.

We use the same set of image augmentations as in BYOL [30] for both BYOL and SimCLR, and also use BYOL's (4096, 256) two-layer projection head for both methods. We follow SimCLR and BYOL to use the LARS optimizer [74] with a cosine decay learning rate schedule [49] over 1000 epochs with a warm-up period, as detailed in Sec. A.4. For ablation experiments we train for 300 epochs instead. As in SimCLR and BYOL, we use batch size of 4096 split over 64 Cloud TPU v3 cores. Except for ablation studies of compression strength, $\beta$ is set to 1.0 for both C-SimCLR and C-BYOL. We follow SimCLR and BYOL in their hyperparameter choices unless otherwise stated, and provide exhaustive details in Sec. A. Pseudocode can be found in Sec. H.

**Evaluation protocol.** We assess the performance of representations pretrained on the ImageNet training set [60] without using any labels. Then we train a linear classifier on different labeled datasets on top of the frozen representation. The final performance metric is the accuracy of these classifiers. As our approach builds on SimCLR [13] and BYOL [30], we follow the same evaluation protocols. Further details are in Sec. B.

## 3.2 Linear Evaluation of Self-supervised Representations

**Linear evaluation on ImageNet.** We first evaluate the representations learned by our models by training a linear classifier on top of frozen features on the ImageNet training set, following standard practice [12, 30, 43, 44]. As shown in Table 1, our compressed objectives provide strong improvements to state-of-the-art SimCLR [12] and BYOL [30] models across different ResNet architectures [34] of varying widths (and thus number of parameters) [75]. Our reproduction of the SimCLR baseline (70.7% top-1 accuracy) outperforms that of the original paper (69.3%). Our implementation of BYOL, which obtains a mean Top-1 accuracy of 74.2% (averaged over three trials) matches that of [30] within a standard deviation.

Current self-supervised methods benefit from longer training schedules [12, 13, 30, 15, 33]. Table 1 shows that our improvements remain consistent for both 300 epochs, and the longer 1000 epoch schedule which achieves the best results. In addition to the Top-1 and Top-5 accuracies, we also compute the Brier score [8] which measures model calibration. Similar to the predictive accuracy, we observe that our compressed models obtain consistent improvements.

**Learning with a few labels on ImageNet.** After self-supervised pretraining on ImageNet, we learn a linear classifier on a small subset (1% or 10%) of the ImageNet training set, using the class labels this time, following the standard protocol of [12, 30]. We expect that with strong feature representations, we should be able to learn an effective classifier with limited training examples.

Table 2 shows that the compressed models once again outperform the SimCLR and BYOL counterparts. The largest improvements are observed in the low-data regime, where we improve upon the state-of-the-art BYOL by 5.1% and SimCLR by 1.8%, when using only 1% of the ImageNet labels. Moreover, note that self-supervised representations significantly outperform a fully-supervised ResNet-50 baseline which overfits significantly in this low-data scenario.

**Comparison to other methods.** Table 3 compares C-SimCLR and C-BYOL to other recent self-supervised methods from the literature (in the standard setting of using two augmented views) on ImageNet linear evaluation accuracy. We present accuracy for models trained for 800 and 1000

Table 1: ImageNet accuracy of linear classifiers trained on representations learned with SimCLR [12] and BYOL [30], with and without CEB compression. A lower Brier score corresponds to better model calibration. We report mean accuracy and standard deviations over three trials.

| | SimCLR | | | BYOL | | |
|---|---|---|---|---|---|---|
| Method | Top-1 | Top-5 | Brier | Top-1 | Top-5 | Brier |
| *ResNet-50, 300 epochs* | | | | | | |
| Uncompressed | 69.1±0.089 | 89.1±0.034 | 42.1±1.06 | 72.8±0.155 | 91.0±0.072 | 37.3±0.089 |
| Compressed | **70.1**±0.177 | **89.6**±0.099 | **41.0**±0.107 | **73.6**±0.039 | **91.5**±0.080 | **36.5**±0.045 |
| *ResNet-50, 1000 epochs* | | | | | | |
| Uncompressed | 70.7±0.094 | 90.1±0.081 | 40.0±0.123 | 74.2±0.139 | 91.7±0.041 | 35.7±0.114 |
| Compressed | **71.6**±0.084 | **90.5**±0.067 | **39.7**±0.876 | **75.6**±0.151 | **92.7**±0.076 | **34.0**±0.127 |
| *ResNet-50 2x, 1000 epochs* | | | | | | |
| Uncompressed | 74.5±0.014 | 92.1±0.031 | 35.2±0.038 | 77.2±0.057 | 93.5±0.036 | 31.8±0.073 |
| Compressed | **75.0**±0.082 | **92.4**±0.086 | **34.7**±0.129 | **78.8**±0.088 | **94.5**±0.016 | **29.8**±0.028 |

Table 2: ImageNet accuracy when training linear classifiers with 1% and 10% of the labeled ImageNet data, on top of frozen, self-supervised representations learned on the entire ImageNet dataset without labels. For the supervised baseline, the whole ResNet-50 network is trained from random initialisation. We report mean results of 3 trials.

| | Top-1 | | Top-5 | |
|---|---|---|---|---|
| Method | 1% | 10% | 1% | 10% |
| Supervised [77] | 25.4 | 56.4 | 48.4 | 80.4 |
| SimCLR | 49.3 | 63.3 | 75.8 | 85.9 |
| C-SimCLR | **51.1** | **64.5** | **77.2** | **86.5** |
| BYOL | 55.5 | 68.2 | 79.7 | 88.4 |
| C-BYOL | **60.6** | **70.5** | **83.4** | **90.0** |

Table 3: Comparison to other methods on ImageNet linear evaluation and supervised baselines. *: trained for 800 epochs. Other methods are 1000 epochs.

| | ResNet-50 | |
|---|---|---|
| Method | 1x | 2x |
| SimCLR [12] | 69.3 | 74.2 |
| SwAV* (2 crop) [11, 15] | 71.8 | - |
| InfoMin Aug* [65] | 73.0 | - |
| Barlow Twins [76] | 73.2 | - |
| BYOL [30] | 74.3 | 77.4 |
| C-SimCLR (ours) | 71.6 | 75.0 |
| C-BYOL (ours) | **75.6** | **78.8** |
| Supervised [12, 30] | 76.5 | 77.8 |

epochs, depending on what the original authors reported. C-BYOL achieves the best results compared to other state-of-the-art methods. Moreover, we can improve C-BYOL with ResNet-50 even further to 76.0 Top-1 accuracy when we train it for 1500 epochs.

**Comparison to supervised baselines.** As shown in Table 3, SimCLR and BYOL use supervised baselines of 76.5% for ResNet-50 and 77.8% for ResNet-50 2x [12, 30] respectively. In comparison, the corresponding compressed BYOL models achieve 76.0% for ResNet-50 and 78.8% for ResNet-50 2x, effectively matching or surpassing reasonable supervised baselines.[4]

The results in Tables 1 to 3 support our hypothesis that compression of SSL techniques can improve their ability to generalize in a variety of settings. These results are consistent with theoretical understandings of the relationship between compression and generalization [61, 68, 22], as are the results in Table 5 that show that performance improves with increasingly strong compression (corresponding to higher values of $\beta$), up to some maximum amount of compression, after which performance degrades again.

### 3.3 Evaluation of Model Robustness and Generalization

In this section, we analyse the robustness of our models to various domain shifts. Concretely, we use the models, with their linear classifier, from the previous experiment, and evaluate them on a suite

---

[4]We note that comparing supervised and self-supervised methods is difficult, as it can only be system-wise. Various complementary techniques can be used to further improve evaluation results in both settings. For example, the appendix of [30] reports that various techniques improve supervised model accuracies, whilst [30, 43] report various techniques to improve evaluation accuracy of self-supervised representations. We omit these in order to follow the common supervised baselines and standard evaluation protocols used in prior work.

Table 4: Evaluation of robustness and generalization of self-supervised models, using a ResNet-50 backbone trained on ImageNet for 1000 epochs. We report the mean Top-1 accuracy over 3 trials on a range of benchmarks (detailed in Sec. 3.3 and the appendix) which measure an ImageNet-trained model's generalization to different domains and distributions.

| Method | ImageNet-A | ImageNet-C | ImageNet-R | ImageNet-v2 | ImageNet-Vid | YouTube-BB | ObjectNet |
|---|---|---|---|---|---|---|---|
| SimCLR | 1.3 | 35.0 | 18.3 | 57.7 | 63.8 | 57.3 | 18.7 |
| C-SimCLR | **1.4** | **36.8** | **19.6** | **58.7** | **64.7** | **59.5** | **20.8** |
| BYOL | 1.6 | 42.7 | 24.4 | 62.1 | 67.9 | 60.7 | 23.4 |
| C-BYOL | **2.3** | **45.1** | **25.8** | **63.9** | **70.8** | **63.6** | **25.5** |

Table 5: Ablation study of CEB compression using C-SimCLR models trained for 300 epochs with a ResNet-50 backbone. $\beta$ controls the level of CEB compression. We evaluate linear-evaluation on ImageNet, and model robustness on the remaining datasets as described in Sec. 3.3.

| Method | ImageNet | ImageNet-A | ImageNet-C | ImageNet-R | ImageNet-v2 | ImageNet-Vid | YouTube-BB | ObjectNet |
|---|---|---|---|---|---|---|---|---|
| SimCLR | 69.0 | **1.2** | 32.9 | 17.8 | 56.0 | 61.1 | 58.3 | 17.6 |
| $\beta = 0$ | 69.7 | 1.1 | 35.8 | 17.6 | 56.8 | 62.5 | 58.4 | 18.5 |
| $\beta = 0.01$ | 69.7 | **1.2** | 36.2 | 17.5 | 57.2 | 61.2 | 58.5 | 18.7 |
| $\beta = 0.1$ | 70.1 | 1.1 | 36.1 | 17.6 | 56.9 | 62.4 | 58.6 | 18.4 |
| $\beta = 1$ | **70.2** | 1.1 | **36.7** | 18.2 | **57.5** | **62.6** | **60.4** | **19.2** |
| $\beta = 1.5$ | 69.7 | 1.1 | 36.4 | **18.3** | 57.3 | 62.0 | 57.9 | 18.5 |

of robustness benchmarks that have the same label set as the ImageNet dataset. We use the public robustness benchmark evaluation code of [19, 18]. As a result, we can evaluate our network and report Top-1 accuracy, as shown in Table 4, without any modifications to the network.

We consider "natural adversarial examples" with ImageNet-A [37] which consists of difficult images which a ResNet-50 classifier failed on. ImageNet-C [37] adds synthetic corruptions to the ImageNet validation set, ImageNet-R [36] considers other naturally occuring distribution changes in image style while ObjectNet [5] presents a more difficult test set for ImageNet where the authors control for different parameters such as viewpoint and background. ImageNet-Vid and YouTube-BB [62] evaluate the robustness of image classifiers to natural perturbations arising in video. Finally, ImageNet-v2 [58] is a new validation set for ImageNet where the authors attempted to replicate the original data collection process. Further details of these robustness benchmarks are in Section F.

Table 4 shows that SimCLR and BYOL models trained with CEB compression consistently out-perform their uncompressed counterparts across all seven robustness benchmarks. This is what we hypothesized in the SimCLR settings based on the Lipschitz continuity argument in Sec. 2.4 and the appendix. All models performed poorly on ImageNet-A, but this is not surprising given that ImageNet-A was collected by [37] according to images that a ResNet-50 classifier trained with full supervision on ImageNet misclassified, and we evaluate with ResNet-50 models too.

## 3.4 The Effect of Compression Strength

Table 5 studies the effect of the CEB compression term, $\beta$ on linear evaluation accuracy on ImageNet, as well as on the same suite of robustness datasets. We observe that $\beta = 0$, which corresponds to no explicit compression, but a stochastic representation, already achieves improvements across all datasets. Further improvements are observed by increasing compression ($\beta$), with $\beta = 1$ obtaining the best results. But overly strong compression can be harmful. Large values of $\beta$ correspond to high levels of compression, and can cause training to collapse, which we observed for $\beta = 2$.

## 4 Related Work

Most methods for learning visual representations without additional annotation can be roughly grouped into three families: generative, discriminative, and bootstrapping. Generative approaches build a latent embedding that models the data distribution, but suffer from the expensive image generation step [69, 59, 28, 39, 42]. While many early discriminative approaches used heuristic pretext tasks [20, 53], multi-view contrastive methods are among the recent state-of-the-art [12, 33, 14, 13, 54, 35, 48, 64, 11].

Some previous contributions in the multi-view contrastive family [65, 76, 63, 21] can be connected to the information bottleneck principle [66, 67, 3] but in a form of unconditional compression as

they are agnostic of the prediction target, i.e. the target view in multiview contrastive learning. As discussed in [26, 27], CEB performs conditional compression that directly optimizes for the information relevant to the task, and is shown theoretically and empirically better [26, 27]. A multi-view self-supervised formulation of CEB, which C-SimCLR can be linked to, was described in [26]. Federici et al. [24] later proposed a practical implementation of that, leveraging either label information or data augmentations. In comparison to [24], we apply our methods with large ResNet models to well-studied large-scale classification datasets like ImageNet and study improvements in robustness and generalization, rather than using two layer MLPs on smaller scale tasks. This shows that compression can still work using state-of-the-art models on challenging tasks. Furthermore, we use the vMF distribution rather than Gaussians in high-dimensional spaces, and extend beyond contrastive learning with C-BYOL.

Among the bootstrapping approaches [31, 10, 30] which BYOL [30] belongs to, BYORL [29] modified BYOL [30] to leverage Projected Gradient Descent [50] to learn a more adversarially robust encoder. The focus is, however, different from ours as we concentrate on improving the generalization gap and robustness to domain shifts.

A variety of theoretical work has established that compressed representations yield improved generalization, including [61, 68, 22]. Our work demonstrates that these results are valid in practice, for important problems like ImageNet, even in the setting of self-supervised learning. Our theoretical analysis linking Lipschitz continuity to compression also gives a different way of viewing the relationship between compression and generalization, since smoother models have been found to generalize better (e.g., [9]). Smoothness is particularly important in the adversarial robustness setting [71, 23, 73], although we do not study that setting in this work.

## 5 Conclusion

We introduced compressed versions of two state-of-the-art self-supervised algorithms, SimCLR [12] and BYOL [30], using the Conditional Entropy Bottleneck (CEB) [27]. Our extensive experiments verified our hypothesis that compressing the information content of self-supervised representations yields consistent improvements in both accuracy and robustness to domain shifts. These findings were consistent for both SimCLR and BYOL across different network backbones, datasets and training schedules. Furthermore, we presented an alternative theoretical explanation of why C-SimCLR models are more robust, in addition to the information-theoretic view [26, 27, 1, 2], by connecting Lipschitz continuity to compression.

**Limitations.** We note that using CEB often requires explicit and restricted distributions. This adds certain constraints on modeling choices. It also requires additional effort to identify or create required random variables, and find appropriate distributions for them. Although we did not need additional trainable parameters for C-SimCLR, we did for C-BYOL, where we added a linear layer to the online encoder, and a 2-layer MLP to create $b(\cdot|y)$. It was, however, not difficult to observe the von Mises-Fisher distribution corresponds to loss function of BYOL and SimCLR, as well as other recent InfoNCE-based contrastive methods [11, 14, 33].

**Potential Negative Societal Impact.** Our work presents self-supervised methods for learning effective and robust visual representations. These representations enable learning visual classifiers with limited data (as shown by our experiments on ImageNet with 1% or 10% training data), and thus facilitates applications in many domains where annotations are expensive or difficult to collect.

Image classification systems are a generic technology with a wide range of potential applications. We are unaware of all potential applications, but are cognizant that each application has its own merits and societal impacts depending on the intentions of the individuals building and using the system. We also note that training datasets contain biases that may render models trained on them unsuitable for certain applications. It is possible that people use classification models (intentionally or not) to make decisions that impact different groups in society differently.

## Acknowledgments and Disclosure of Funding

We thank Toby Boyd for his help in making our implementation open source.

John Canny is also affiliated with the University of California, Berkeley.

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
