# Appendices

We provide additional implementation details in Section A, details of our linear evaluation protocol in Section B, experiments on transfer of the compressive representations to other classification tasks in Section C, and further ablations in Section E. We provide further details about our robustness evaluation in Section F. Finally, we provide a more detailed explanation of the relation between Lipschitz continuity and SimCLR with CEB compression, introduced in Section 2.4 of the main paper, in Section G.

## A   Implementation details and hyperparameters

In this section, we further describe our implementation details. Our implementation is based off of the public implementation of SimCLR [12]. In general, we closely follow the design choices of BYOL [30] for both of our SimCLR and BYOL implementations. Despite having different objectives, BYOL and SimCLR share many components, including image augmentations, network architectures, and optimization settings. As explained in the original paper [30], BYOL itself may be considered as a modification to SimCLR with a slow moving average target network, an additional predictor network, and switching the learning target from InfoNCE to a regression loss. Therefore, many of the design choices and hyperparameters are applicable to both. As explained in Section 3.1, we align SimCLR with BYOL on the choices of image augmentations, network architecture, and optimization settings in order to reduce the number of variables in comparison.

### A.1   Image augmentations

During self-supervised training, we use the set of image augmentations from BYOL [30] for all our models.

- Random cropping: randomly select a patch of the image, with an area uniformly sampled between 8% and 100% of that original image, and an aspect ratio logarithmically sampled between $3/4$ and $4/3$. Then this patch is resized to $224 \times 224$ using bicubic interpolation.

- Left-to-right flipping: randomly flip the patch.

- Color jittering: brightness, contrast, saturation and hue of an image are shifted by a uniformly random offset. The order to apply these adjustments is randomly selected for each patch.

- Color dropping: RGB pixel values of an image are converted to grayscale according to $0.2989r + 0.5870g + 0.1140b$.

- Gaussian blurring: We use a $23 \times 23$ kernel to blur the $224 \times 224$ image, with a standard deviation uniformly sampled over $[0.1, 2.0]$.

- Solarization: This is a color transformation $x = x \cdot \mathbf{1}_{\{x<0.5\}} + (1 - x) \cdot \mathbf{1}_{\{x \geq 0.5\}}$ for pixels with values in $[0, 1]$ (we convert all pixel values into floats between $[0, 1]$).

As described in Sec. 2, we use augmentation functions $t$ and $t'$ to transform an image into two views. $t$ and $t'$ are compositions of the above image augmentations in the listed order, each applied with a predefined probability. The image augmentation parameters to generate $t$ and $t'$ are listed in Table 6.

During evaluation, we perform center cropping, as done in [12, 30]. Images are resized to 256 pixels along the shorter side, after which a $224 \times 224$ center crop is applied. During both training and evaluation, we normalize image RGB values by subtracting the average color and dividing by the standard deviation, computed on ImageNet, after applying the augmentations.

**Differences from the original SimCLR [12].**   Since the image augmentation parameters that BYOL [30] uses are different from the original SimCLR, we list the original SimCLR parameters in the last column of Table 6, which are the same for $t$ and $t'$, to clarify the differences. Additionally, the original SimCLR samples the aspect ratio of cropped patches uniformly, instead of logarithmically, between $[3/4, 4/3]$.

Table 6: Image augmentation parameters. We use the hyperparameter values from BYOL [30], and include the values from the original SimCLR [12] as reference.

| Parameter | $t$ | $t'$ | Orig. SimCLR [12] |
|---|---|---|---|
| Random crop probability | 1.0 | 1.0 | 1.0 |
| Flip probability | 0.5 | 0.5 | 0.5 |
| Color jittering probability | 0.8 | 0.8 | 0.8 |
| Brightness adjustment max strength | 0.4 | 0.4 | 0.8 |
| Contrast adjustment max strength | 0.4 | 0.4 | 0.8 |
| Saturation adjustment max strength | 0.2 | 0.2 | 0.8 |
| Hue adjustment max strength | 0.1 | 0.1 | 0.2 |
| Color dropping probability | 0.2 | 0.2 | 0.2 |
| Gaussian blurring probability | 1.0 | 0.1 | 1.0 |
| Solarization probability | 0.0 | 0.2 | 0.0 |

## A.2 Network architecture

Following [12, 30], we use ResNet-50 [34] as our backbone convolutional encoder (the "Conv" part in Figure 1 and Figure 2). We vary the ResNet width [75] (and thus the number of parameters) from 1× to 2×. In Sec. D, we additionally report C-BYOL results with different ResNet depth, from 50 to 152. The representations $h_x, h_y$ in SimCLR and $h, h_t, h'_t$ in BYOL correspond to the 2048-dimensional (for ResNet-50 1×) final average pooling layer output. These representations are projected to a smaller space by an MLP (called "projection" in Figure 1 and Figure 2). As in [30], this MLP consists of a linear layer with output size 4096 followed by batch normalization [41], ReLU, and a final linear layer with output dimension 256. $q(\cdot)$ in BYOL/C-BYOL (Fig. 2) is called the predictor. The predictor $q(\cdot)$ is also a two-layer MLP which shares the same architecture with the projection MLP [30].

**Differences from the original SimCLR [12].** The original SimCLR [12] uses a 2048-d hidden layer and a 128-d output layer for the projection MLP, after which an additional batch normalization is applied to the 128-d output. Both BYOL [30] and our work do not have this batch normalization on the last layer. We did not observe significant change in performance for the uncompressed models and found it harmful to the compressed models.

## A.3 von Mises-Fisher Distributions

We use the vMF implementation in public Tensorflow Probability (TFP) library [17], specifically the current TFP version 0.13. We have found that sampling and computing log probabilities in high dimensions with the current TFP version has become sufficiently stable and fast to train all of the models in our paper.[5]

## A.4 Optimization

We follow BYOL [30] for our optimization settings. During self-supervised training, we use the LARS optimizer [74] with a cosine decay learning rate schedule [49] over 1000 epochs and a linear warm-up period at the beginning of training. The linear warm-up period is 10 epochs in most cases. We increase it to 20 epochs for BYOL and C-BYOL with larger ResNets (ResNet-50 2x, ResNet-101, ResNet-152) as we found it helpful to prevent mode collapse and improve performance. In most cases, we set the base learning rate to 0.2 and scale it linearly by batch size (LearningRate = 0.2 × BatchSize/256). For C-BYOL, we increase the base learning rate to 0.26 for better performance. For careful comparison, we extensively searched base learning rate for BYOL but did not find a configuration better than 0.2 as used in the original work [30]. We use a weight decay of $1.5 \times 10^{-6}$. For the BYOL/C-BYOL target network, the exponential moving average update rate $\alpha$ starts from

---

[5]Previous versions of TFP were unstable for sampling from vMF distributions with higher than 5 dimensions, and at the time of writing, the authors of the library have not updated the documentation to indicate that this is no longer the case.

Table 7: Ablation study on BYOL models trained for 300 epochs. Top-1 denotes the linear evaluation Top-1 accuracy on ImageNet.

| Method | Top-1 |
|---|---|
| BYOL $w_{\text{byol}} = 1.0$ | 72.5 |
| BYOL $w_{\text{byol}} = 5.0$ | 72.8 |
| BYOL $w_{\text{byol}} = 5.0$ + 256-d linear layer + $l_2$-normalization | 72.8 |
| BYOL $w_{\text{byol}} = 5.0$ + 256-d linear layer + $l_2$-normalization + sampling | 72.8 |
| C-BYOL $w_{\text{byol}} = 5.0$ | 73.6 |

Table 8: The effect of loss weights on BYOL models trained for 1000 epochs. Top-1 denotes the linear evaluation Top-1 accuracy on ImageNet.

| Method | Top-1 |
|---|---|
| BYOL $w_{\text{byol}} = 1.0$ | 74.2 |
| BYOL $w_{\text{byol}} = 2.0$ | 74.2 |
| BYOL $w_{\text{byol}} = 5.0$ | 74.2 |

$\alpha_{\text{base}} = 0.996$ and ramps up to 1 with a cosine schedule, $\alpha \triangleq 1 - (1 - \alpha_{\text{base}})(\cos(\pi k/K) + 1)/2$ where $k$ is the current training step and $K$ is the total number of training steps.

For 300-epoch models used in ablations, we set the base learning rate to 0.3 in most cases, and increase it to 0.35 for C-BYOL. We use a weight decay of $10^{-6}$. For BYOL and C-BYOL, the base exponential moving average update rate $\alpha_{\text{base}}$ is set to 0.99.

We note that there is a small chance that both BYOL and C-BYOL can end up with collapsed solutions, but it mostly happens in early phase of training and is easy to observe with the learning objective spiking or reaching NaN.

**Differences from the original SimCLR [12].**    Optimization settings of the original SimCLR are very similar but, for 1000-epoch training, they use a base learning rate of 0.3 and weight decay of $10^{-6}$.

## A.5   SimCLR and C-SimCLR details

As described in Section A.1, Section A.2, Section A.4, we made minor modifications to the original SimCLR to align with BYOL on the choices of image augmentations, network architecture, and optimization settings. With these modifications, our SimCLR baseline reproduction outperforms the original (top-1 accuracy 70.6% versus 69.3%).

For C-SimCLR, we use $\kappa_e = 1024$ for the true encoder $e(\cdot|x)$ and $\kappa_b = 10$ for the backward encoder, where $e(\cdot|x)$ and $b(\cdot|y)$ are von Mises-Fisher distributions. The compression factor $\beta$ that we use for C-SimCLR is 1.0. Note that the original SimCLR has temperature $\tau = 0.1$ which is equivalent to having $\kappa_b = 10$, since $\kappa_b = 1/\tau$.

## A.6   BYOL and C-BYOL details

As shown in Table 7 and Table 8, our BYOL implementation stably reproduces results comparable to [30] with 300 and 1000 epochs of training. An interesting behavior we observed is that, for shorter training with 300 epochs, scaling the BYOL regression loss can improve performance. Specifically we add a weight multiplier $w_{\text{byol}} = \kappa_d/2$ to the BYOL loss Eq. (13).

$$L_{\text{byol}} = w_{\text{byol}}||\mu_e - y'||_2^2 \tag{23}$$

Table 7 shows that multiplying the loss by five increases the linear evaluation accuracy from 72.5% to 72.8%. This improvement is consistent across multiple runs. Therefore, we choose $w_{\text{byol}} = 5$ for 300-epoch BYOL/C-BYOL. However, we do not see the same improvement for 1000-epoch models. Table 8 shows that $w_{\text{byol}}$ makes little difference for 1000-epoch BYOL models. We still choose $w_{\text{byol}} = 2$ for all 1000-epoch BYOL and C-BYOL models since it tends to work better than $w_{\text{byol}} = 1$ for the compressed models and models with larger ResNets.

Table 9: Transfer to other classification tasks, by performing linear evaluation. The backbone network is ResNet-50, pretrained in a self-supervised fashion for 1000 epochs.

| Method | Food101 | CIFAR10 | CIFAR100 | Flowers | Pet | Cars | Caltech-101 | DTD | SUN397 | Aircraft | Birdsnap |
|--------|---------|---------|----------|---------|------|------|-------------|------|--------|----------|----------|
| SimCLR | 72.5 | 91.1 | 74.4 | 88.4 | 83.5 | 49.7 | 89.5 | 72.5 | 61.8 | 51.6 | 35.4 |
| C-SimCLR | **73.0** | **91.6** | **75.2** | **89.0** | **84.0** | **52.7** | **91.2** | **73.0** | **62.3** | **53.5** | **38.2** |

Furthermore, Table 7 verifies that the additional linear layer with $l_2$-normalization that we added for C-BYOL and $z$ sampling (both were described in Section 2.3) do not result in a difference in performance. The improvement happens only when CEB compression is used.

We set $\kappa_e = 16384.0$, $\kappa_b = 10.0$, and the compression factor $beta = 1.0$ for C-BYOL if not specified otherwise.

# B  Linear evaluation protocol on ImageNet

As common in self-supervised learning literature [30, 12, 43, 14], we assess the performance of our representations learned on the ImageNet training set (without labels) by training a linear classifier on top of the frozen representations using the labeled data. For training this linear classifier, we only apply the random cropping and flipping image augmentations. We optimize the cross-entropy loss using SGD with Nesterov momentum over 40 epochs. We use a batch size of 1024 and momentum of 0.9 without weight decay, and sweep the base learning rate over $\{0.4, 0.3, 0.2, 0.1, 0.05\}$ to choose the best learning rate on a validation set, following [30]. We perform center cropping during evaluation, as done in [12, 30]. Images are resized to 256 pixels along the shorter side, after which the $224 \times 224$ center crop is selected. During both training and evaluation, we normalize image RGB values by subtracting the average color and dividing by the standard deviation, computed on ImageNet, after applying the augmentations.

**Learning with a few labels**    In Section 3.2 we described learning the linear classifier on 1% and 10% of the ImageNet training set with labels. We performed this experiment with the same 1% and 10% splits from [12].

# C  Transfer to other classification tasks

We analyze the effect of compression on transfer to other classification tasks in Table 9. This allows us to assess whether the compressive representations learned by our method are generic and transfer across image domains.

**Datasets.**    We perform the transfer experiments on the Food-101 dataset [7], CIFAR-10 and CIFAR-100 [46], Birdsnap [6], SUN397 [72], Stanford Cars [45], FGVC Aircraft [51], the Describable Texture Dataset (DTD) [16], Oxford-IIIT Pets [55], Caltech-101 [25], and Oxford 102 Flowers [52]. We carefully follow their evaluation protocol, i.e. we report top-1 accuracy for Food-101, CIFAR-10, CIFAR-100, Birdsnap, SUN397, Stanford Cars, anad DTD; mean per-class accuracy for FGVC Aircraft, Oxford-IIIT Pets, Caltech-101, and Oxford 102 Flowers. These datasets are also used by [12, 30, 44]. More exhaustive details about train, validation, and test splits of these datasets can be found in Section D of [30] (arXiv v3).

**Transfer via linear classifier.**    To demonstrate the effectiveness of compressed representations, we compare SimCLR and C-SimCLR representations as an example. We freeze the representations of our model and train an $\ell_2$-regularized multinomial logitstic regression classifier on top of these frozen representations. We minimize the cross-entropy objective using the L-BFGS optimizer. As in [30, 12], we selected the $\ell_2$ regularization parameter from a range of 45 logarithmically spaced values between $[10^{-6}, 10^5]$.

We observe in Table 9 that our Compressed SimCLR model consistently outperforms the uncompressed SimCLR baseline on each of the 11 datasets we tested. We note absolute improvements in accuracy ranging from 0.5% (CIFAR-10, SUN397) to 3% (Stanford Cars). These experiments

Table 10: C-BYOL and BYOL trained for 1000 epochs with different ResNet depth. We report ImageNet Top-1 accuracy from linear evaluation, averaged over 3 trials.

| | C-BYOL | | BYOL [30] | |
| Architecture | Top-1 | Top-5 | Top-1 | Top-5 |
|---|---|---|---|---|
| ResNet-50 | **75.6** | **92.7** | 74.3 | 91.6 |
| ResNet-101 | **77.8** | **93.9** | 76.4 | 93.0 |
| ResNet-152 | **78.7** | **94.4** | 77.3 | 93.7 |

suggest that the representations learned by compressed model are generic, and transfer beyond the ImageNet domain which they were learned on.

## D Extra C-BYOL results with Deeper ResNets

In Table 10, we additionally report results of C-BYOL and BYOL retrained for 1000 epochs with ResNet-101 and ResNet-152, as it could be of interest to demonstrate improvements over the state-of-the-art BYOL on these deeper ResNet models. It can be observed that C-BYOL gives significant gains across ResNets with different depths.

## E Additional Ablations

The hyperparameter and architecture choices of SimCLR and BYOL have been investigated in the original works [12, 30]. Here we focus on analysing hyperparameters specific to C-SimCLR and C-BYOL.

Table 11, Table 12 and Table 13 show how changing $\kappa_e$ and $\kappa_b$ affect the results, respectively. We also investigate the effect of the compression factor $\beta$ for C-BYOL (Table 14) in addition to the compression analysis for SimCLR in Sec. 3.4. Similar to C-SimCLR, as compression strength ($\beta$) increases, the linear evaluation result improves, with $\beta = 1.0$ obtaining the best results, but overly strong compression leads to a drop in performance.

Finally, we conduct a preliminary exploration on the interplay between CEB compression and image augmentations, using cropping area ratio as an example in Table 15. As described in Section A.1, we follow [30, 12] to randomly crop an image to an area between 8% and 100% of the original image. We refer to this 8% as the "area lower bound", which is the most aggressive cropping area ratio that can happen. As the area lower bound decreases, we are reducing the amount of information that can be shared between the two representations, because there is less and less mutual information between the two images: $I(X; X')$ gets smaller the more we reduce the area lower bound [65]. Thus, smaller area lower bounds should force the model to be more compressed. What we see in Table 15 is that the SimCLR models are much more sensitive to the changes in the area lower bound than the C-SimCLR models are. We speculate that this is because the compression done by the C-SimCLR objective overlaps to some extent with the compression given by varying the area lower bound. Regardless, the compression due to the area lower bound hyperparameter appears to be insufficient to adequately compress away irrelevant information in the SimCLR model, which is why the C-SimCLR models continue to outperform the SimCLR models at all area lower bound values.

## F Robustness benchmark details

In this section, we provide some additional details on each of the datasets used in our robustness evaluations. Note that we use the public robustness benchmark evaluation code of [19, 18].[6]

ImageNet-A [38]: This dataset of "Natural adversarial examples" consists of images of ImageNet classes which a ResNet-50 classifier failed on. The dataset authors performed manual, human-verification to ensure that the predictions of the ResNet-50 model were indeed incorrect and egregious [38].

---

[6]https://github.com/google-research/robustness_metrics

Table 11: The effect of varying $\kappa_e$ for C-SimCLR models. We report ImageNet Top-1 accuracy from linear evaluation.

| $\kappa_e$ | 256 | 512 | 1024 | 2048 | 4096 | 8192 |
|---|---|---|---|---|---|---|
| ImageNet Top-1 accuracy | 69.8 | 69.8 | 70.2 | 69.8 | 69.6 | 69.6 |

Table 12: The effect of varying $\kappa_e$ for C-BYOL models. We report ImageNet Top-1 accuracy from linear evaluation.

| $\kappa_e$ | 4096 | 8192 | 16384 | 32768 |
|---|---|---|---|---|
| ImageNet Top-1 accuracy | 73.0 | 73.3 | 73.6 | 73.2 |

Table 13: The effect of varying $\kappa_b$ for C-SimCLR and C-BYOL models. We report ImageNet Top-1 accuracy from linear evaluation.

| Method | $\kappa_b =1$ | 3 | 10 | 15 | 20 |
|---|---|---|---|---|---|
| C-SimCLR | 65.0 | 68.5 | 70.2 | 69.1 | 68.6 |
| C-BYOL | 73.1 | 73.3 | 73.6 | 73.4 | 73.2 |

Table 14: The effect of $\beta$ on C-BYOL. Note that Table 5 in the main paper studied this effect on C-SimCLR. The final column is the uncompressed BYOL baseline.

| $\beta$ | 1.5 | 1.0 | 0.1 | 0.01 | BYOL |
|---|---|---|---|---|---|
| ImageNet Top-1 accuracy | 73.4 | 73.6 | 73.1 | 73.0 | 72.8 |

Table 15: The effect of varying the area range lower rounds for SimCLR and Compressed SimCLR. We report the ImageNet Top-1 accuracy from linear evaluation. Note how the baseline SimCLR model is much more sensitive to this data-augmentation hyperparameter.

| Method | 8% | 16% | 25% | 50% |
|---|---|---|---|---|
| SimCLR | 69.0 | 68.6 | 67.6 | 61.4 |
| C–SimCLR | 70.2 | 70.0 | 68.9 | 64.3 |

ImageNet-C [37]: This dataset adds 15 corruptions to ImageNet images, each at 5 levels of severity. We report the average accuracy over all the corruptions and severity levels.

ImageNet-R [36]: This dataset, which has the full name "Imagenet Rendition", captures naturally occuring distribution changes in image style, camera operation and geographic location.

ImageNet-v2 [58]: This is a new test set for ImageNet, and was collected following the same protocol as the original ImageNet dataset. The authors posit that the collected images are more "difficult", and observed consistent accuracy drops across a wide range of models trained on the original ImageNet.

ObjectNet [5]: This is a more challenging test set for ImageNet, where the authors control for different viewpoints, backgrounds and rotations. Note that ObjectNet has a vocabulary 313 object classes, of which 113 are common with ImageNet. Following [18], we evaluate on only the images in the dataset which have one of the 113 ImageNet labels. Our network is still able to predict any one of the 1000 ImageNet classes though.

ImageNet-Vid and YouTube-BB [62] evaluate the robustness of image classifiers to natural perturbations arising in video. This dataset was created by [62] by augmenting the ImageNet-Vid [60] and YouTube-BB [57] datasets with additional annotations.

# G   Analysis of Lipschitz Continuity and Compression

In this section, we provide a more detailed explanation of the relation between Lipschitz continuity and SimCLR with CEB compression, introduced in Section 2.4. Lipschitz Continuity provides a way of measuring how smooth a function is. For some function $f$ and a distance measure $D(f(x_1), f(x_2))$,

Lipschitz continuity defines an upper bound on how quickly $f$ can change as $x$ changes:

$$L||\Delta x|| \geq D(f(x), f(x + \Delta x)) \tag{24}$$

where $L$ is the Lipschitz constant, $\Delta x$ is the vector change in $x$, and $||\Delta x|| > 0$.

Frequently, the choice of $D$ is the absolute difference function: $|f(x_1) - f(x_2)|$. However, we can use a multiplicative distance rather than an additive distance by considering the absolute difference of the logs of the functions:

$$D(f(x_1), f(x_2)) \equiv |\log f(x_1) - \log f(x_2)| \tag{25}$$

It is trivial to see that Equation (25) obeys the triangle inequality, which can be written:

$$|a - b| \geq |a| - |b| \tag{26}$$

Equation (26) is true for any scalars $a$ and $b$. Setting $a = \log f(x_1)$ and $b = \log f(x_2)$ is sufficient, given that $f(\cdot)$ is a positive, scalar-valued function. For $D(\cdot)$ to be a valid distance metric, $f(x)$ must also satisfy the identity of indiscernibles requirement: $f(x_1) = f(x_2) \Leftrightarrow x_1 = x_2$. If that requirement is violated, then $D(\cdot)$ becomes a pseudometric, which is inconsistent with Lipschitz continuity.

Noting that $|a - b| \equiv \max(a - b, b - a)$, we will simplify the analysis by considering the two arguments to the implicit $\max$ in Equation (25) one at a time, starting with:

$$L \geq \frac{1}{||\Delta x||}(\log f(x) - \log f(x + \Delta x)) \tag{27}$$

If we define $f(x)$ to be our encoder distribution, $e(z|x)$, we get a function of $z$ of Lipschitz value:[7]

$$L(z) \geq \frac{1}{||\Delta x||}(\log e(z|x) - \log e(z|x + \Delta x)) \tag{28}$$

Note that the encoder distribution must not violate the identity of indiscernibles property: $\forall z : e(z|x_1) = e(z|x_2) \Leftrightarrow x_1 = x_2$. This is not the case in general, but for reasonable distribution families, the sets of $z$ that violate this property for any $(x_1, x_2)$ pair will have measure zero. In the case that $e(z|\cdot)$ is parameterized by some function $f(\cdot)$, such as a neural network, $f$ must also not violate the identity of indiscernibles property. This argues in favor of using invertible networks for $f$, or at least not using activation functions like relu that are likely to cause $f$ to map multiple $x$ values to some constant. We note that in practice, it doesn't seem to matter, as shown empirically in Section 3.

As $e(z|x)$ is parameterized by the output of our model, Equation (28) captures the semantically relevant smoothness of the model. For example, if our encoder distribution is a Gaussian with learned mean and variance, the impact of the model parameters on the means is semantically distinct from the impact of the model parameters on the variance. In that setting, using the parameter vectors themselves naively in a Lipschitz formulation like $L_{\text{naive}}||\Delta x|| \geq ||f_\theta(x) - f_\theta(x + \Delta x)||$, where $f_\theta$ outputs concatenated mean and variance parameters, would clearly fail to correctly capture the model's smoothness. Our formulation using the encoder *distribution* directly does not have this flaw, and thus generalizes to capture a notion of smoothness for any choice of distribution parameterization. Note that this notion of smoothness of the distribution still depends directly on the the smoothness of the underlying function that generates the distribution's parameters, while also capturing the smoothness of the distribution itself.

We can remove the dependence on $z$ of Equation (28) by taking the expectation over $z$ with respect to $e(z|x)$. This gives us an *expected* Lipschitz value:

$$\mathbb{E}_{z \sim e(z|x)} L(z) \geq \frac{1}{||\Delta x||} \mathbb{E}_{z \sim e(z|x)} \log e(z|x) - \log e(z|x + \Delta x) \tag{29}$$

$$= \frac{1}{||\Delta x||} \text{KL}[e(z|x)||e(z|x + \Delta x)] \tag{30}$$

---

[7]Note that if we choose an encoder distribution where the density ever goes to 0 or $\infty$, Equation (28) will have a maximum value of $\infty$. Of course, it's generally easy to avoid this situation by choosing "well-behaved" distributions like the Gaussian or von Mises-Fisher distributions, whose densities are non-zero on the entire domain, and to parameterize them with variance or concentration parameters that don't go to 0 or $\infty$, respectively.

It is important to note that Equation (30) no longer obeys the triangle inequality, due to the KL divergence, since it is easy to find three distributions $p, q, r$ such that $\text{KL}[p||q] > \text{KL}[p||r] + \text{KL}[q||r]$.

We could also have computed the expectation over $z \sim e(z|x + \Delta x)$, yielding:

$$\mathbb{E}_{z\sim e(z|x+\Delta x)}L(z) \geq -\frac{1}{||\Delta x||}\,\text{KL}[e(z|x + \Delta x)||e(z|x)] \tag{31}$$

But this is trivially true due to $L(z)$ being non-negative and the negative KL term being non-positive, so we can ignore this term here. However, when we consider the second argument to the implicit $\max$ in Equation (25), the negative and positive KL terms are swapped, and we are left with:

$$\mathbb{E}_{z\sim e(z|x+\Delta x)}L(z) \geq \frac{1}{||\Delta x||}\,\text{KL}[e(z|x + \Delta x)||e(z|x)] \tag{32}$$

When we take the expectations over $z$, the resulting KL divergences have an underlying quadratic growth in $||\Delta x||$: as $||\Delta x||$ increases linearly, the KL divergences increase quadratically.[8] This is why the KL divergence violates the triangle inequality, and also why it is problematic for measuring Lipschitz continuity: in general, $L$ will be unbounded when measured by the KL even when the underlying function $f(x)$ parameterizing the distributions has a bounded Lipschitz constant, since the KL will always grow faster then $||\Delta x||$. We can address this by instead considering the squared Lipschitz constant:

$$L^2||\Delta x||^2 \geq \text{KL}[e(z|x)||e(z|x + \Delta x)] \quad \text{and} \quad L^2||\Delta x||^2 \geq \text{KL}[e(z|x + \Delta x)||e(z|x)] \tag{33}$$

which is equivalent to:

$$L^2 \geq \frac{1}{||\Delta x||}\mathbb{E}_{z\sim e(z|x)}L(z) \quad \text{and} \quad L^2 \geq \frac{1}{||\Delta x||}\mathbb{E}_{z\sim e(z|x+\Delta x)}L(z) \tag{34}$$

Finally, we note the following relationship:

$$L^2 = \max_{x,\Delta x} \max\left( \frac{1}{||\Delta x||^2}\,\text{KL}[e(z|x)||e(z|x + \Delta x)], \frac{1}{||\Delta x||^2}\,\text{KL}[e(z|x + \Delta x)||e(z|x)] \right) \tag{35}$$

In words, the true squared Lipschitz constant of the encoder is equal to the least smooth $(x, \Delta x)$ pair, as measured by the greater of the two KL divergences at that pair.

Putting all of this together, we observe that the following two KL divergences together give a lower bound on the encoder's Lipschitz constant:

$$L^2 \geq \frac{1}{||\Delta x||^2} \max\left( \text{KL}[e(z|x)||e(z|x + \Delta x)], \text{KL}[e(z|x + \Delta x)||e(z|x)] \right) \tag{36}$$

Thus, taking the pointwise maximum across pairs of inputs in any dataset gives a valid estimate of the maximum lower bound of the encoder's Lipschitz constant. Equation (36) can be evaluated directly on any pair of valid inputs $(x, x + \Delta x)$. Equation (36) is the same as Equation (20) used in Section 2.4.

**Example: the von Mises-Fisher distribution.** An exponential family distribution has the form:

$$h(z)\exp(\eta^T T(z) - A(\eta)) \tag{37}$$

where $T(z)$ is the sufficient statistic, $\eta$ is the canonical parameter, and $A(\eta)$ is the cumulant. For the von Mises-Fisher distribution, which has the form:

$$C_n(\kappa)\exp(\kappa\mu^T z) \tag{38}$$

we have $h(z) = 1$, $T(z) = z$ and $A(\eta)$ is the negative log of the normalizing constant $C_n(\kappa)$. Instead of a general parameter vector $\eta$, the standard von Mises-Fisher distribution uses a unit vector $\mu = \eta/||\eta||$ and a scale or concentration parameter $\kappa = ||\eta||$.

---

[8]This is easiest to see with Gaussian distributions whose means are parameterized by an identity map of $x$ and $x + \Delta x$: the KL divergence is quadratic in difference of the means, which is $||\Delta x||$.

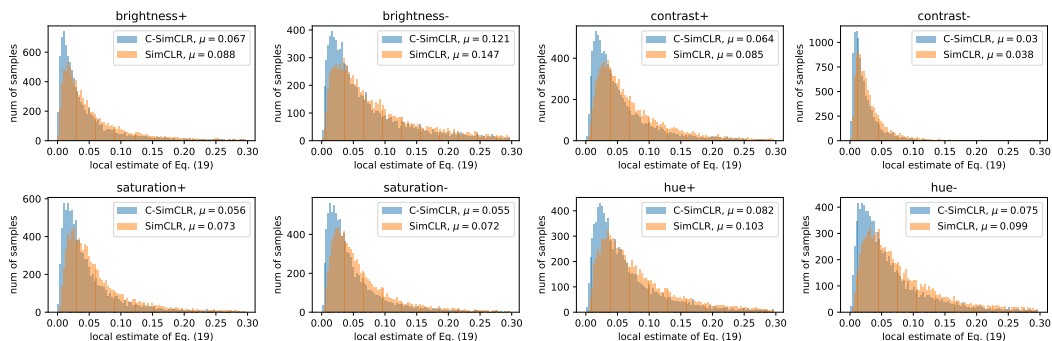

Figure 3: Histograms of Equation (19) (also Equation (28) in this section) on 10,000 training images. Each local estimate is Equation (28) with a $(x, x + \Delta x)$ pair. Here $x$ is the original image and $x + \Delta x$ is the augmented image. SimCLR is in orange. C-SimCLR is in blue. Higher $y$-axis values for lower $x$-axis values are better. We also report the mean ($\mu$) values. C-SimCLR consistently outperforms SimCLR.

If $e(z|x)$ is parameterized by a deterministic neural network for the von Mises-Fisher canonical parameter denoted $\overline{e}(x)$, then we have:

$$e(z|x) = C_n(||\overline{e}(x)||) \exp(\overline{e}(x)^T z) \tag{39}$$

and $\mathrm{KL}[e(z|x)||e(z|y)]$ (define $y = x + \Delta x$) is:

$$(\overline{e}(x) - \overline{e}(y))^T \overline{z}(x) + \log C_n(||\overline{e}(x)||) - \log C_n(||\overline{e}(y)||) \tag{40}$$

where $\overline{z}(x)$ is the mean direction function of the distribution ($\overline{e}(x) = ||\overline{e}(x)||\overline{z}(x)$). The symmetric KL-divergence ($\mathrm{KL}[e(z|x)||e(z|y)] + \mathrm{KL}[e(z|y)||e(z|x)]$) is then:

$$(\overline{e}(x) - \overline{e}(y))^T(\overline{z}(x) - \overline{z}(y)) \tag{41}$$

which is closely related to the $L_2^2$ norm of the vector $\overline{e}(x) - \overline{e}(y)$.

Furthermore, we can choose $\kappa = ||\overline{e}(x)||$ as a hyperparameter and just parameterize $e(z|x)$'s unit length mean direction $\overline{z}(x)$. Apart from choosing different $\kappa$ hyperparameters, this is exactly what we do in the C-SimCLR setting described in Section 2.2.

Specifically, in Section 2.2, minimizing the residual information term $I(X; Z|Y)$ correspond to minimizing $\mathrm{KL}[e(z|x)||b(z|y)]$ instead of $\mathrm{KL}[e(z|x)||e(z|y)]$, where $b$ and $e$ have the same mean direction parameterization but different $\kappa$ hyperparameters, say $\kappa_e$ and $\kappa_b$. We can show that the two KLs are actually consistent as learning objectives. With $\kappa_e, \kappa_b$ as hyperparameters, $\mathrm{KL}[e(z|x)||e(z|y)]$ (Equation (40)) can be written as

$$\kappa_e(\overline{z}(x) - \overline{z}(y))^T \overline{z}(x) + \log C_n(\kappa_e) - \log C_n(\kappa_e), \tag{42}$$

and $\mathrm{KL}[e(z|x)||b(z|y)]$ can be written as

$$(\kappa_e \overline{z}(x) - \kappa_b \overline{z}(y))^T \overline{z}(x) + \log C_n(\kappa_e) - \log C_n(\kappa_b) \tag{43}$$

$$= (\kappa_e - \kappa_b) + \kappa_b(\overline{z}(x) - \overline{z}(y))^T \overline{z}(x) + \log C_n(\kappa_e) - \log C_n(\kappa_b). \tag{44}$$

It is not difficult to see that the two KLs are only different in scale and by a constant, and thus are consistent as learning objectives. As we claimed in Section 2.4 (after Equation (22)), the use of different constant hyperparameters $\kappa_e$ and $\kappa_b$ in the encoders of $x$ and $y$ only changes the minimum achievable KL divergences. We can reach the same conclusion for the residual information in another direction $I(Y; Z|X)$. Thus, whether or not $\kappa_e$ and $\kappa_b$ are the same, we are still minimizing the Lipschitz constant of our encoder function at each observed $(x, y)$ pair when we minimize the residual information terms in the bidirectional CEB objective (Equation (12)).

**Estimating the local Lipschitz constant.** We can evaluate Equation (19) (also Equation (28) on any $(x, x + \Delta x)$ pairs to estimate how smooth our model is at that point, and to compare the relative smoothness of different models. Here, we consider $(x, x + \Delta x)$ pairs where $x$ is taken either from

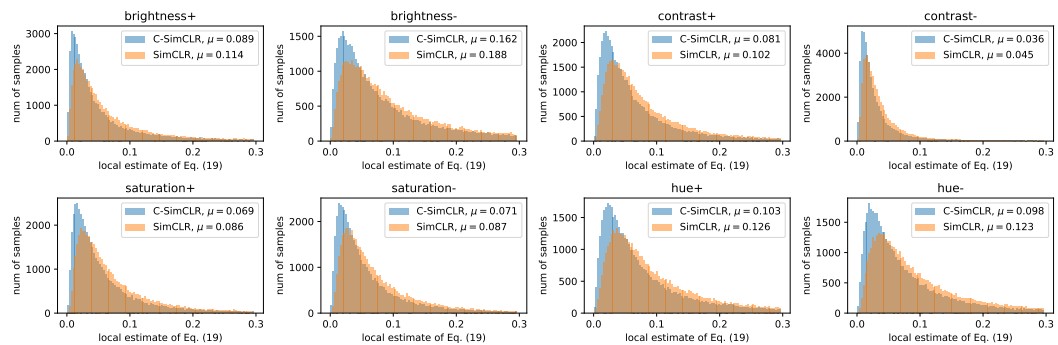

Figure 4: The same as Figure 3, but on 50,000 validation images.

the training or the validation set (using only a center crop in both cases), and $x + \Delta x$ is generated by either increasing or decreasing exactly one of: brightness, contrast, saturation, or hue. The absolute changes are the maximum adjustment strength in our image defined in Table 6 (e.g. for increasing brightness, we increase by 0.4). In Figures 3 and 4, we compare the histograms of Equation (19) on the SimCLR ResNet-50 model and the corresponding C-SimCLR ResNet-50 model. On both datasets and all eight augmentations, the C-SimCLR models have substantially more mass in the lower values of the local Lipschitz estimates for those image pairs, and have lower mean values computed over the dataset. Additionally, the mean C-SimCLR results on the *validation* set are almost all lower or equal to the mean SimCLR results on the *training* set, so the smoothness improvements from adding compression to SimCLR appear to be substantial. The only exceptions are for 'brightness+' (SimCLR training mean: 0.088, C-SimCLR validation mean: 0.089) and 'brightness-' (SimCLR training mean: 0.147, C-SimCLR validation mean: 0.162).

## H   Pseudocode

Listings 1 and 2 show Tensorflow pseudocode for C-SimCLR and C-BYOL respectively.

```python
tfd = tensorflow_probability.distributions

def simclr_ceb_loss(x,
                    y,
                    f_enc,
                    kappa_e=1024.0,
                    kappa_b=10.0,
                    beta=1.0):
  """Compute a Contrastive version of CEB loss for C-SimCLR model.

  In practice, we follow SimCLR to apply this loss in a bidirectional manner as
  loss = simclr_ceb_loss(x, y) + simclr_ceb_loss(y, x).
  We use the same notation as the main paper.

  Args:
    x: An augmented image view. The expected shape is [B, H, W, C].
    y: An augmented image view. The expected shape is [B, H, W, C].
    f_enc: An image encoder (Conv + Projection in Fig. 1).
    kappa_e: A float. Concentration parameter of distribution e.
    kappa_b: A float. Concentration parameter of distribution b.
    beta: CEB beta for controlling compression strength (Equation 1).

  Returns:
    A tensor `loss`. The loss is per-sample.
  """
  # Obtain unit-length mean direction vectors with expected shape [B, r_dim].
  r_x = tf.math.l2_normalize(f_enc(x), -1)
  r_y = tf.math.l2_normalize(f_enc(y), -1)

  batch_size = tf.shape(r_x)[0]
  labels_idx = tf.range(batch_size)
  # Labels are pseudo-labels which mark corresponding positives in a batch
  labels = tf.one_hot(labels_idx, batch_size)
  mi_upper_bound = tf.math.log(tf.cast(batch_size, tf.float32))

  e_zx = tfd.VonMisesFisher(r_x, kappa_e)
  b_zy = tfd.VonMisesFisher(r_y, kappa_b)
  z = e_zx.sample()
  log_e_zx = e_zx.log_prob(z)
  log_b_zy = b_zy.log_prob(z)
  i_xzy = log_e_zx - log_b_zy   # residual information I(X;Z|Y)
  logits_ab = b_zy.log_prob(z[:, None, :])   # broadcast

  # The following categorical corresponds to c(y|z) and d(x|z) in Equation 12:
  cat_dist_ab = tfd.Categorical(logits=logits_ab)
  h_yz = -cat_dist_ab.log_prob(labels_idx)
  i_yz = mi_upper_bound - h_yz
  loss = beta * i_xzy - i_yz

  return loss
```

Listing 1: Tensorflow pseudocode of C-SimCLR.

```python
tfd = tensorflow_probability.distributions

def byol_ceb_loss(x,
                  x_prime,
                  f_enc,
                  f_enc_target,
                  q_net,
                  l_net,
                  m_net,
                  kappa_e=16384.0,
                  kappa_b=10.0,
                  beta=1.0,
                  byol_loss_weight=2.0):
  """Compute loss for C-BYOL model.

  The notation corresponds to Section 2.3 and Figure 2 of the paper.
  This code presents an updated version of C-BYOL as described in the
  general response.

  In practice, we follow BYOL to apply this loss in a bidirectional manner as
  loss = byol_ceb_loss(x, x_prime, ...) + byol_ceb_loss(x_prime, x, ...).
  We use the same notation as the main paper.

  Args:
    x: An augmented image view. The expected shape is [B, H, W, C].
    x_prime: An augmented image view. The expected shape is [B, H, W, C].
    f_enc: An image encoder (Conv + Projection in Fig. 2).
    f_enc_target: The target image encoder. A slow moving-average of f_enc.
    q_net: The BYOL predictor, which is a two-layer MLP.
    l_net: A transformation function. We choose a linear layer in this work.
    m_net: A transformation function. We choose a two-layer MLP in this work.
    kappa_e: A float. Concentration parameter of distribution e.
    kappa_b: A float. Concentration parameter of distribution b.
    beta: CEB beta for controlling compression strength (Equation 1).
    byol_loss_weight: BYOL loss weight. byol_loss_weight = kappa_d/2.

  Returns:
    A tensor `loss`. The loss is per-sample.
  """
  r = f_enc(x)
  mu_e = tf.math.l2_normalize(q_net(r), -1)
  e_zx = tfd.VonMisesFisher(mu_e, kappa_e)
  z = e_zx.sample()
  y_pred = tf.math.l2_normalize(l_net(z), -1)

  r_t = tf.math.l2_normalize(f_enc_target(x), -1)
  y = tf.stop_gradient(r_t)
  mu_b = tf.math.l2_normalize(m_net(y), -1)
  b_zy = tfd.VonMisesFisher(mu_b, kappa_b)

  r_t_prime = tf.math.l2_normalize(f_enc_target(x_prime), -1)
  y_prime = tf.stop_gradient(r_t_prime)

  # byol_loss corresponds to -log d(y|z) as described in Section 2.3
  byol_loss = tf.reduce_sum(tf.math.square(y_pred - y_prime), axis=-1)

  log_e_zx = e_zx.log_prob(z)
  log_b_zy = b_zy.log_prob(z)
  i_xzy = log_e_zx - log_b_zy

  loss = byol_loss_weight * byol_loss + beta * i_xzy

  return loss
```

Listing 2: Tensorflow pseudocode of C-BYOL.