# OpenReview forum: "Compressive Visual Representations"
_NeurIPS.cc/2021/Conference — NeurIPS 2021 Poster_

### Official Review · Reviewer_1tWc · 2021-07-14

**Rating:** 7
**Confidence:** 4

**Summary:**

This work adds a conditional entropy bottleneck (CEB) objective to two existing frameworks for self-supervised learning, namely SimCLR and BYOL. The motivation for adding CEB is to encourage information compression in the representations learned from these models which can make the representations more robust. The authors measure the performance of SimCLR and BYOL with added CEB objective empirically on standard benchmarks such as linear evaluation protocol on ImageNet with different amounts of training data. They provide evidence that adding CEB outperforms other existing self-supervised learning models on these benchmarks, including vanilla SimCLR and BYOL.

**Limitations And Societal Impact:**

Yes, they have.

**Main Review:**

### Strengths
- While the information theoretic CEB objective is not novel, to the best of my knowledge, incorporating it into SimCLR and BYOL has not been done before and is a well-motivated idea. This approach can be applied in other settings.
- The experimental setup is exhaustive. Мodels with the proposed modified objective are evaluated on popular benchmarks (including the standard ImageNet linear classification protocol and linear classification on versions of ImageNet designed to evaluate robustness and generalization) and show strong empirical performance when compared to other existing methods.
- The paper provides a link between Lipschitz continuity and CEB compression in the case of SimCLR.
- This is a well written paper and it is organized clearly.

###Questions & Comments
1. How many seeds were used for results in Tables 2-5? What was the standard deviation?
2. What is the full range of values that were used for $\beta$?
3. I think it would be beneficial for the main text to clearly indicate the best values for $\beta$ for both SimCLR and BYOL in all the different experiments.

###Nitpicks
- Equation (6): $z_x$ and $z_y$ should be replaced by $r_x$ and $r_y$, the sum in the denominator should be over lower case $k$ (not capital $K$)
- Line 155: in “distribution of ||z||” should use LaTeX, $\||z\||$


**Time Spent Reviewing:**

8

---

> ### Author Response · Authors · 2021-08-10
> **Response to Reviewer 1tWc**
>
> **How many seeds were used for results in Tables 2-5? What was the standard deviation?**
>
> Our results in Tables 1 and 3 use 3 seeds. Table 2, 4, 5 use 1 seed. We will update the results with averages and standard deviations over 3 runs for Tables 2 and 4. The average over multiple runs for Tables 2 and 4 are still consistently better than the baselines. Due to computational requirements, we don’t plan on doing multiple runs for the ablation study in Table 5 (six rows x three trials = 18 experiments).
>
>
> **What is the full range of values that were used for β? Indicate the best values of β for all experiments?**
>
> The $\beta$ values that we have studied are presented in Table 5 (for C-SimCLR) and Table 12 (for C-BYOL). Except for these two tables we use one fixed value of $\beta = 1.0$ for C-SimCLR and $\beta = 0.01$ for C-BYOL. These hyperparameter values for C-SimCLR and C-BYOL were listed in Sec. A.4 and Sec. A.5 respectively, but we will move these to the main paper.
>
> **Nitpicks**
> Thank you. We will correct these in the revision.

---

> > ### Comment · Reviewer_1tWc · 2021-08-22
> > **Thank you!**
> >
> > Thank you for your response! I am keeping my original score and look forward to the updated version of the paper.

---

### Official Review · Reviewer_ZEAE · 2021-07-15

**Rating:** 7
**Confidence:** 5

**Summary:**

The paper shows how to use the Conditional Entropy Bottleneck (CEB)[1] in a self-supervised fashion.  Thus resulting in a new method for training approximately minimal self-supervised  (SSL) representations, i.e., a new member of the SSL information bottleneck family (although extremely similar to [2]). Specifically, they show how to use CEB in conjunction with SimCLR and BYOL. Furthermore, they provide good arguments to suggest that their proposed objective will learn smoother encoders than standard self-supervised models, which can help robustness. Through extensive experiments,  they show significant performance gains of the proposed methods compared to SimCLR and BYOL.

- [1] Fischer, Ian. "The conditional entropy bottleneck." Entropy 22.9 (2020): 999.
- [2] Federici, Marco, et al. "Learning robust representations via multi-view information bottleneck." arXiv preprint arXiv:2002.07017 (2020).

**Limitations And Societal Impact:**

The limitation and societal impact are generally well discussed. Although, as previously stated, I would like to see a comparison (and thus potential limitations of the current method) compared to previous SSL + IB papers.

**Main Review:**

**Post-rebuttal**: I've updated my score given that the authors (seem to) have addressed all my major concerns (related work / mathematical mistakes / practical details) in the updated manuscript.

----------

**Overall**: This paper has interesting ideas with strong experimental results. It will likely be of interest to the SSL and IB sub-communities. I nevertheless do not think that the current version is ready for publication given that (1) it fails to discuss very related work; (2) has mathematical mistakes; (3) it's very hard to understand what authors do in practice. Fortunately, all these problems can be easily solved and I am happy to increase my score if the authors make the necessary changes.

**Strengths**:
- **Significant empirical gains and thorough experimental section**. The experimental results show significant gains against very strong baselines. These results are even more impressive given the thorough evaluation protocol.
- **Interest to the community**: The link between the proposed CEB and smoothness of the learned objective is interesting and to my knowledge novel. Such a smoothness argument might be useful beyond robustness. Furthermore, although SSL+IB has already been considered (see "related work" below), the final objective used with vMF variational family is novel and has many potential advantages compared to previous work (which I hope the authors will discuss in the final version).
- **Thorough appendices**: The supplemental material is thorough and very useful.

**Weaknesses**:
- **Missing important related work**: The discussion of related work is insufficient. Most importantly, multi-view IB [1] provides a loss that is extremely similar to your C-simlcr objective (the only real difference seems to be the variational family) and provides experiments in robustness and standard SSL just as you. More generally, the paper reads as if it is the first (although the authors do not explicitly say so)  to investigate the use of "compressed"/"low information" representations in SSL, which is definitely not the case. The paper also fails to cite previous work that uses/notice the link between von Mises-Fisher distribution and contrastive learning with normalized representations. This definitely diminishes the novelty of the paper, but I do not find that to be an issue as long as it is correctly discussed. Authors should take advantage of their "related work section" to discuss the advantages/limits of the proposed extension of CEB compared to previous work. More details can be found in my "related work" section below.
- **Some mathematical mistakes/imprecisions** currently there are many mathematical issues that obscure some parts/equations of the paper. The biggest issues are the following (see "suggestions" for other issues):
     -  Equation (5) has many mistakes, to the point that I don't even know what you were trying to show and how many mistakes there are. The mistakes that are obvious are: (1) what do you mean by lower bounding a conditional entropy by a distribution? surely you meant $E[-\log d(y|z)]$; (2) you say that you drop $H[Y]$ because that's a constant, but that's only true if you have some minimization, do you mean taking min on both sides? (3) you say that this is the InfoNCE bound but you are removing $\frac{1}{k}$ from the denominator, is that on purpose? If so say that it's InfoNCE up to $log K$. I actually think there are deeper issues, but I'm not even sure what you aiming for due to the previous issues. Here's what I think you meant by using InfoNCE bound: $I[Z,Y] \geq I_{NCE}$ so $H[Y] - I[Z,Y] = H[Y|Z]  \leq H[Y] - I_{NCE}$ so $\inf_{e(z|x)} H[Y|Z]  \leq \inf_{e(z|x)} - I_{NCE} =  \inf_{e(z|x)} - \log K - E[ \log d(y|z)]$ so $\inf_{e(z|x)} H[Y|Z]  \leq \inf_{e(z|x)} E[- \log d(y|z)]$. But if that is the case you are: (1) missing a negative sign; (2) used lower bound instead of an upper bound; (3) should really just use the standard cross-entropy bound instead of InfoNCE to get the same result.
    - I think that the “distance" measure that you are using is not an actual metric (it’s only a pseudo metric) because it does not satisfy $d(x_1,x_2) = 0 \iff x_1=x_2$, indeed, you can have $e(z|x_2) = e(z|x_1)$ for different $x_1,x_2$.  As a result, you cannot formally talk about Lipchitz continuity and the results from the paper you cite in line 191 might not hold. I would suggest adding a footnote explaining that you are actually using a pseudometric so you are not exactly considering Lipchitz continuity but hypothesize that this provides similar robustness guarantees. I actually don’t think that it makes much difference but this has to be flagged.
- **Hard to understand what is done in practice and missing important implementation details**: it's actually quite hard to get what is done in practice because there are many moving components and these are scattered in different places of the paper (e.g. (5) and (11) are the equations that are needed for C-SimCLR but separated by much text and equations that is not used for the model). It would help a lot to have clear algorithms for C-SimCLR and C-BYOL. Other non-trivial and important details are missing, e.g., how you sample (with gradient flows) from the vMF which is known to be very hard in high dimensions.

**Related work**
- **Multi-View Information Bottleneck**: As previously stated [1] is extremely similar to your paper on many fronts. As a result,  I think that this paper should be cited in your introduction where you should explain what you are contributing which they didn't. Furthermore, the exact difference needs to be discussed in the text.
- **Self supervised learning + information bottleneck**:  besides [1] there are other (albeit less related) papers that consider "compressed"/"low information" representations in SSL. For example, Barlow Twins[2] can be seen from an SSL IB point of view  (see their Appx. A).  [3] provides some theory for this type of SSL+IB. [4] provides different losses for learning compressed SSL representations (although this seems to be online since March so contemporary work).  The only similar work that you cite is InfoMin Aug, which arguably is the least related.
- **Von Mises-Fisher + cosine similarity**: the relation between vMF and contrastive loss with cosine similarity has already been pointed out/used by others, e.g., [5-7]. Please add some discussion about those or at least acknowledge some of them around line 113.
- **Minimality and generalization**. Your experimental results clearly show that compressed / low-information representations help downstream performance. There’s a large body of work that proves/suggests that this will be the case and I think that these should be cited.   E.g., [8] proves that a minimal representation ensures a generalization gap of 0 (i.e. tight but only for optimal representations), while [9,10] provides upper bounds (although loose) on the generalization gap based on $I[Z, X]$.

**Suggestions / specific issues:**
- I would say very early on that you are conditioning on a second view instead of the label (as usual in CEB). This was not clear to me when reading until I arrived at section 2.2.
- The use and meaning of $r_x,r_y$ vs $z_x,z_y$ is unclear. Can you please clarify around lines 102-103 how $r$ relates to $z$? From section 2.2. it seems that $r$ refers to parameters of the density of $z$ but this should be clearer much sooner. Also, in equation (6) should it be $L_{nce}(r_x,r_y)$?
- I find section 2.3 hard to follow. I would suggest simplifying this section, although to be honest, I’m not sure how, because there are many moving components in BYOL.
- Section 3.2. is missing some motivation/explanation as to why compression should help in standard iid setting  (i.e. it’s missing a paragraph like at lines 271-273 for robustness). As I said in “related work” I would cite the body of work that proves [8] or suggests [9,10] that low information representations provide strong guarantees on generalization to give an intuition as to why this makes sense and why you consider the experimental setting 3.2.

**Questions**
- Right now in equation (11) it seems that you are estimating the KL of the vMF distributions using samples, have you considered using an upper bound on the KL between vMF such as in [11]? This would essentially maximize $r_x^Tr_y$ which is much simpler and already symmetric.
- How do you sample in practice from the vMF with a reparametrization trick in high dimension? Implementations of vMF usually suffer from numerical issues as the Bessel function is very hard to evaluate in high dimensions (e.g. TensorFlow's implementation does not work for more than 5 dimensions).  E.g., [12] had to come up with a clever parametrization trick for vMF, is that what you are using?

**Minor Suggestions**
- Equation (3): this should be an equal sign instead of define
- Line 81: say that d(y|z) is also a variational approximation
- Line 82: bounded -> upper-bounded
- Line 88: cannot take gradients -> cannot easily take gradients
- Line 107: $r_{y_k}$ -> $r_y$
- Line 107: corresponds to $b(z|y)$ ->corresponds to the unnormalized  $b(z|y)$       (or use $\frac{1}{C}exp$instead of exp, where $C$ is normalising constant).
- Line 114: n-dimensional -> (n-1)-dimensional     (or change pdf to n+1 in 115)
- Line 123: “relax” makes it sound as if the resulting $I_{NCE}$ is not a proper lower bound anymore, but that’s not the case, right? If it’s still a lower bound I’d use “extend”.
- Equation 15: I don’t think the first equality is correct: both densities have different domains and even different graphs. I think you mean that $e(z|x)=e(w \cdot \phi |x)$ where  $e(w ,\phi |x) = e(w | x) \cdot e( \phi |x)$.
- Line 158: “and since C is a constant.” I don’t understand what you mean here?
- Line 161: Continuity -> continuity
- Equation 18: say that L is maxed over L(z)
- Line 175: isn’t -> is not
- I would replace all $\min$ with $\inf$ (e.g. equation 4,9,10,11)

**References:**
- [1] Federici, Marco, et al. "Learning robust representations via multi-view information bottleneck." arXiv preprint arXiv:2002.07017 (2020).
- [2] Zbontar, Jure, et al. "Barlow twins: self-supervised learning via redundancy reduction." arXiv preprint arXiv:2103.03230 (2021).
- [3] Sridharan, Karthik, and Sham M. Kakade. "An information-theoretic framework for multi-view learning." (2008): 403.
- [4] Dubois, Yann, et al. "Lossy Compression for Lossless Prediction." arXiv preprint arXiv:2106.10800 (2021).
- [5] Wang, Tongzhou, and Phillip Isola. "Understanding contrastive representation learning through alignment and uniformity on the hypersphere." International Conference on Machine Learning. PMLR, 2020.
- [6] Hasnat, Md, et al. "von mises-fisher mixture model-based deep learning: Application to face verification." arXiv preprint arXiv:1706.04264 (2017).
- [7] Kumar, Sachin, and Yulia Tsvetkov. "Von mises-fisher loss for training sequence to sequence models with continuous outputs." arXiv preprint arXiv:1812.04616 (2018).
- [8] Dubois, Yann, et al. "Learning optimal representations with the decodable information bottleneck." arXiv preprint arXiv:2009.12789 (2020).
- [9] Shamir, Ohad, Sivan Sabato, and Naftali Tishby. "Learning and generalization with the information bottleneck." International Conference on Algorithmic Learning Theory. Springer, Berlin, Heidelberg, 2008.
- [10] Vera, Matías, Pablo Piantanida, and Leonardo Rey Vega. "The role of information complexity and randomization in representation learning." arXiv preprint arXiv:1802.05355 (2018).
- [11] Diethe, Tom. "A Note on the Kullback-Leibler Divergence for the von Mises-Fisher distribution." arXiv preprint arXiv:1502.07104 (2015).
- [12] Davidson, Tim R., et al. "Hyperspherical variational auto-encoders." arXiv preprint arXiv:1804.00891 (2018).

**Time Spent Reviewing:**

15 hours

---

> ### Author Response · Authors · 2021-08-10
> **Response to Reviewer ZAEA**
>
> **References:**
>
> Thank you for the list of related work. We will add all requested papers to the related work section.
>
>
> * **The multi-view IB paper (Federici et al. 2020):**
> Thanks for suggesting this work. For prior work of C-SimCLR style multi-view SSL using CEB, in our paper we refer to the CEB paper (Fischer, 2018 [17]), which describes SSL in its Sec. C.3. The multi-view IB paper proposed a practical implementation that leverages either label information or data augmentations. We will thoroughly discuss the difference between the multi-view IB paper and ours in the revisions. Major differences between the multi-view IB paper and our own include:
>
>   - On the theoretical side, we note the relationship between bidirectional CEB and the Lipschitz constant, and their relevance to the symmetric SimCLR setting, helping to explain why compressed models give improvements to generalization and robustness.
>   - To validate our method we apply it to well-studied large-scale classification datasets like ImageNet, where we can also study and compare improvements in robustness and generalization.
>   - We use larger image models (ResNet50 and ResNet50x2), rather than the two layer MLPs used on the tasks in the multi-view IB paper, which shows that compression can still work using state-of-the-art models on challenging tasks like ImageNet.
>   - We use the vMF distribution rather than Gaussians. We additionally show that both sampling and computing log probabilities of vMFs works in high dimensions in practice.
>   - We show that CEB works even in BYOL’s regression setting (we are unaware of any previous published work showing that CEB models work on large scale regression tasks).
>
> * **Self supervised learning + information bottleneck:**
> Thanks for the suggested work. We will add discussion of them in the revisions.
>
> * **Von Mises-Fisher + Cosine similarity:**
> Thanks for these suggested references. We will include them in revisions.
>
> * **Minimality and generalization theory results:**
> All the compression/generalization theory results we’re aware of rely on discrete distributions, so they don't automatically apply to the continuous representation setting we are using. (For example, Shamir, Ohad, Sivan Sabato, and Naftali Tishby. "Learning and generalization with the information bottleneck." International Conference on Algorithmic Learning Theory. Springer, Berlin, Heidelberg, 2008 is a completely trivial upper bound in the continuous setting, due to the reliance on $|T|$, the cardinality of the representation.) We will check whether that’s the case for the references you found that we weren’t already familiar with, and we will include more discussion of this in the revision.
>
> **Equation 5:**
>
> Thanks for raising this concern, Equation 5 should have been written:
>
> $H(Y|Z) \leq E_{x,y \sim p(x,y), z \sim e(z|x)} \log \frac{b(z|y)}{\sum_{k=1}^K b(z|y_k)}$
>
> As you say, this only differs from InfoNCE by a constant $log K$, which we will make more explicit in the text. This does yield a cross entropy loss, as you suggest, but in exactly the same way that InfoNCE is a cross entropy loss -- the cross entropy is using a categorical distribution across minibatch indices, rather than requiring a tractable distribution over the observations in the minibatch. (In a standard classification problem, $Y$ is a class label, so the cross entropy loss only needs a categorical distribution, but for high dimensional $Y$, such as images, the cross entropy loss would normally require a tractable conditional distribution over images).
>
> Separately we should have noted that $d(y|z) \equiv \frac{b(z|y)}{\sum_{k=1}^K b(z|y_k)}$ is a valid variational approximation of the true but unknown $p(y|z)$. This fact is pointed out in the Entropy version of the CEB paper in equations 27 through 29.
>
> **Distance measure:**
>
> Thank you for the suggestion.
>
> We are aware that the KL divergence does not obey the triangle inequality. It is an oversight that we don’t make that explicit in the paper. We will clarify that in the revision.
>
> We would also like to note that it is actually not possible to have e(z|x) = 0 or e(z|y) = 0 in our setting, as we mention on line 165. So long as the distribution has unbounded support (e.g. Gaussian or vMF distributions), the density is non-zero everywhere and it prevents the KL from violating the identity requirement of a distance metric.
>
> **Hard to understand what is done in practice and missing important implementation details:**
>
> Your suggestion to add explicit algorithms to the paper is a good one. We will include them in the updated paper. Note that given the proper formulation they are simple modifications to the SimCLR and BYOL algorithms -- the main salient differences are parameterizing the vMF distributions in the correct places, sampling from the distributions, and computed log probabilities of those samples to compute the additional CEB loss terms.
>
> It’s an oversight on our part that we did not mention that we are sampling from vMF distributions using the public Tensorflow Probability (TFP) library, specifically the current TFP version 0.13. We have found that sampling and computing log probabilities in high dimensions with the current TFP version is sufficiently stable and fast to train all of the models in our paper. Previous versions of TFP were unstable for sampling from vMF distributions with higher than 5 dimensions, and unfortunately, the authors of the library have not updated the documentation to say that this is no longer the case.
>
> **Suggestions / specific issues:**
> * **Conditioning on a second view:**
> We will clarify that in the revisions.
>
>
> * **Meaning of $r_x$, $r_y$ vs $z_x$, $z_y$ is unclear:**
> $r_x$ is the mean parameter for $e(z|x)$. In the bidirectional setting as in standard SimCLR, $r_x$ is the mean parameter for both $e(z_x|x)$ and $b(z_y|x)$. We will clarify in the revisions. Eq (6) should be $L_{nce}(r_x, r_y)$.
>
>
> * **Section 2.3 difficult to follow:**
> We will improve Sec. 2.3 in the revision. One important simplification that we describe in our general response is that we were able to make C-BYOL work without the compound random variable, replacing vMF + Gaussian with a single vMF. Thus, some of the complexity of describing C-BYOL in section 2.3 will be removed (equations 13-15 will be replaced with one equation, and lines 152-157 will be substantially simplified).
>
> * **Explanation of results in Section 3.2:**
> Thank you for the suggestion -- we will add a paragraph that makes it more clear that all of these results are consistent with the original hypothesis that compression of SSL techniques can improve generalization, and that generally results improve with stronger compression (higher beta), up to some maximum.
>
> **Questions:**
> * **Have you considered using analytical KL upper bound instead?**
> This is a design choice. Since Monte Carlo estimates are simple, fast, and tractable, we prefer them in our implementation. Using an analytical KL upper bound is compatible with the CEB objective, however, so it would be a reasonable choice.
> * **How do you sample in practice from the vMF with a reparametrization trick in high dimension?**
> [This has been answered above in our response to “Hard to understand what is done in practice and missing important implementation details.”]

---

> > ### Comment · Reviewer_ZEAE · 2021-08-18
> > **Further clarifications**
> >
> > Thank you for all your responses. I have some important comments about them:
> >
> > **Distance measure**
> > I think I wasn't clear about this. My point had nothing to do with KL not satisfying the triangular inequality. In appendix F you define the "distance" measure to be $d(x_1,x_2) = | \log e(z|x_1)  - \log e(z|x_2)| $. This function is symmetric and satisfies the triangle inequality but it is *not* a metric because it does not satisfy $d(x_1,x_2) = 0 \iff x_1 = x_2$. Actually, many inputs could be mapped to the same distribution so the identity of the indiscernible would hold for no z in sampled from e(z|x). And this has nothing to do with the boundedness of the support of the distribution you are using.
> >
> > As a result, it is not correct to talk about Lipschitz continuity. As previously said I don't think it changes much but this should definitely be explicitly mentioned in the main paper: your objectives are not about Lipschitz continuity but something defined in a similar way for pseudo-metrics. One implication is that formal claims in other papers relating generalization and Lipschitz continuity might not hold for you (but in any case no formal proofs were pointed to, and the relation between Lipschitz constant and robustness in the cited papers seem to be more on the intuitive level).
> >
> > **References**
> > - **multi-view IB**: I agree that the practical difference is the scale + the variational family. It would actually have been great if you could have compared to this as a baseline, to understand whether your very good performance comes from the use of a different variational family, or simply due to scaling up. Of course, I also understand that you already have more experiments than what is needed to have a good paper.
> > - **Minimality and generalization**: I do not completely agree about your point on continuous settings, as in reality,  everything is finite on a computer due to floating-point precisions. That being said I agree that [9,10] are actually uninteresting in practice as they depend on the cardinality $|\mathcal{Z}|$ which is very large even on practical computers. [8] is actually very different as it shows that you get optimal generalization gap when $I[Z,X]$ is minimal, i.e., there are no vacuous bounds. That being said their result only holds at optima, which of course will not be achieved in practice. Nevertheless, I still think these 3 papers do point out the link between generalization and compression and think that mentioning it in Section 3.2 would really make the motivation/results from that section more clear.
> >
> >
> > **Minor**
> > - equation 5: Thank you for correcting these mistakes, this now makes sense.
> > - thank you for providing the TensorFlow code, having an algorithm/pseudo code seemed to be something all reviewers wanted.

---

> > > ### Author Response · Authors · 2021-08-19
> > > **Response**
> > >
> > > **Distance measure**
> > >
> > > Thank you for the clarification. Any choice of $f(x)$ that violates the identity of indiscernibles will have that problem. For example, if $f(x)$ is a ReLU network, it may violate the requirement. We additionally assume that the distributional families being chosen maintain identity of indiscernibles so long as their parameters (which are directly tied to $f(x)$) maintain that property, up to some measure zero set of $z$. We will make that clear in the revision with the following sentences:
> > > * For $D(\cdot)$ to be a valid distance metric, $f(x)$ must also satisfy the identity of indiscernibles requirements: $f(x_1) = f(x_2)  \Leftrightarrow x_1 = x_2$. If that requirement is violated, then $D(\cdot)$ becomes a pseudometric, which is inconsistent with Lipschitz continuity.
> > > * Also note that the encoder distribution must not violate the identity of indiscernibles property: $\forall z: e(z|x_1) = e(z|x_2) \Leftrightarrow e(z|f(x_1)) = e(z|f(x_2)) \Leftrightarrow x_1 = x_2$. We note that this is not the case in general, but that for reasonable distribution families, the sets of $z$ that violate this property for any $(x_1, x_2)$ pair will have measure zero, so we do not expect to encounter such $z$ by sampling.
> > >
> > > **References**
> > > * **Multi-view IB:**
> > > The choice of using vMF in C-SimCLR corresponds to our SimCLR baseline’s heuristic of L2 normalizing the representations. Initially, we experimented with Gaussians, but found that it was unable to outperform the SimCLR baseline, and sometimes unstable to train. (We hypothesized that the instability was due to the unbounded nature of the representation -- we can always move $e(z|x_i)$ further away from $e(z|x_j)$ by making the means larger in some dimension but having opposite signs.)
> > >
> > >   Also, we note that choosing Gaussian corresponds to using L2-distance instead of cosine similarity in the deterministic case. The SimCLR paper didn’t provide this ablation, but the BYOL paper showed that using L2-distance results in a performance drop in their table 20, although the problem setting is different from SimCLR and multi-view IB. We agree that a careful ablation of the choice of distribution would benefit the paper, although space limitations and computational limitations may prevent us from adding such an ablation in revision.
> > >
> > > * **Minimality and generalization:**
> > > Thanks for the additional comments on [8]. We will certainly discuss [8, 9, 10] in the revisions.

---

> > > > ### Comment · Reviewer_ZEAE · 2021-08-19
> > > > **I've updated my score**
> > > >
> > > > Thank you for the detailed answer.
> > > >
> > > > I've updated my score given that all my initial concerns (related work / mathematical mistakes / practical algorithms) seem to have been addressed in the updated manuscript. I think this is now a good paper and I'm looking forward to reading the final version.

---

### Official Review · Reviewer_ofpQ · 2021-07-16

**Rating:** 6
**Confidence:** 3

**Summary:**

Contrastive learning approaches learn salient representations that will be effective for various downstream tasks, while not all learned information is relevant to the task target. Given this insight, the authors aim to learn compressed representations by adding information compression objectives to existing algorithms.

They applied conditional entropy bottleneck (CEB) objective to SimCLR and BYOL to compress away the irrelevant information from the learned representations. In addition, they discovered that enforcing the information compression entails the encoder with bounded Lipschitz constant which is closely related to robustness. Extensive experiments confirm their hypotheses that adding compression regularizer can improve the performance in terms of accuracy and robustness to domain shift.

**Limitations And Societal Impact:**

the authors have adequately addressed the limitations and potential negative societal impact of their work

**Main Review:**

They present their work with reasonable motivations. Overall, I think this paper would be interesting to the NeruIPS audience.

I have some technical concerns:
- It is unclear to me what are the architectures and outputs of decoders $d(.), c(.)$. Are they supposed to predict the inputs $x, y$?
- Is it possible to insert decoders $d(.), c(.)$ into Figure 1&2 just like you did for $e(.), b(.)$. I believe that would help people to get a better understanding of your work.
- I like the diagrams in Figure 1&2. But I would suggest to spend a few sentences to explicitly explain the work flows. For example, I think you can make them more friendly for readers if you can explain what are the red dash lines implying.
- Although the absolute difference is small, I would suggest to report both mean and standard deviation. For instance, in table 1, the last these accuracies are just marginally better than baselines, if std dev is available that would help review to clearly see the significance of experimental results.

**Time Spent Reviewing:**

12

---

> ### Author Response · Authors · 2021-08-10
> **Response to Reviewer ofpQ**
>
> **It is unclear to me what are the architectures and outputs of decoders d(.), c(.). Are they supposed to predict the inputs x, y?**
>
> We will update Sec. 2.2 and 2.3 in the revision to clarify this: $c(.)$ and $d(.)$ are density functions of $x$ and $y$ conditioned on $z_x$ and $z_y$ respectively. In C-SimCLR, we don’t explicitly instantiate predictions of x and y, but use InfoNCE to approximate $c(.)$ and $d(.)$, as described in Sec. 2.1 and 2.2. In other words, we are only using distributions over the latent variable, $z$, rather than distributions over the inputs $x$ and $y$.
>
> In C-BYOL (Sec. 2.3), we parameterized $d(.)$ as a fully connected layer which predicts y from z. Note that in C-BYOL, $y$ refers to a vector which is a function of $x’$, whereas in C-SimCLR, y refers to one of the augmentations of the input.
>
> Please also refer to the relevant Tensorflow pseudo-code of C-SimCLR and C-BYOL in our response to reviewer GtXm, which explicitly clarifies what the $c(.)$ and $d(.)$ distributions correspond to.
>
> **Improvements to figures**
>
> Thank you. We will improve Figures 1 and 2 as advised and add further descriptions to the captions.
>
> **Having standard deviations in table 1**
>
> Thank you. Please refer to the general response where we provided standard deviations for the results in Table 1 and also achieved further improvements with C-BYOL.

---

> > ### Comment · Reviewer_ofpQ · 2021-08-23
> > **Thanks for the clarifications**
> >
> > Thanks for the detailed reply. The clarifications were helpful. Looking forward to read the final version.

---

### Official Review · Reviewer_GtXm · 2021-07-16

**Rating:** 7
**Confidence:** 3

**Summary:**

The paper posits that compressing the representation during learning improves the robustness of said representation. To demonstrate their argument, they derive an objective for SimCLR and BYOL that explicitly compresses the representation using the Conditional Entropy Bottleneck. The addition of this compression improves the performance for both BYOL and SimCLR on ImageNET. They also demonstrate that their method improves the robustness of the encoder and its representation to perturbation of the input data.

**Limitations And Societal Impact:**

The authors discuss some limitations of their work.

The authors present a generic discussion of the societal impacts.

**Main Review:**

The paper presents a principled and novel approach to learning a compressed representation in the context of contrastive methods. While the method is well detailed for both SimCLR and BYOL, it could be helpful to have a pseudo-code of the objective function detailing how the computation is done in practice. Other than that, the paper is very well written. The argument is concise and easy to follow.

The experiments convincingly show the benefits of the approach proposed. While the authors demonstrate that their method performs better than the state-of-the-art of ImageNet, they also provide an ablation demonstrating that the improvement is due to their modifications. They also demonstrate that their method improves robustness on various benchmarks.

Overall, the paper provides a good demonstration that the compression of latent representations improves robustness.

### Question / comments
* L107: I am confused about the correspondance between $exp(\dfrac{1}{\tau} r_{yk}^\top r_x)$ and $b(z|y)$. Isn't $y_k$ the *negative* views and $y$ the *positive* view. I.e., shouldn't the former be a correspondance with $b(z|y_k)$?
* While the authors claim that they will release their code upon acceptance, they could have released an anonymized version.
* The practical realization of $e$, $b$, $c$ and $d$ were not clear to me. The confusion could be resolved by presenting a pseudo-code of the practical implementation of the method.
* It is not clear to me why a model Lipschitz constant generally relates to model robustness. The authors present some works that relate "types of model robustness" to the model's Lipschitz constant. What are the types of model robustness? How do these works relate to your model?

**Time Spent Reviewing:**

6

---

> ### Author Response · Authors · 2021-08-10
> **Response to Reviewer GtXm**
>
> **Notation around L107**
>
> Thank you for spotting this. We will correct it in the revision. $\exp(\frac{1}{\tau} r_y^Tr_x)$ corresponds to $b(z|y)$ and $\exp(\frac{1}{\tau} r_{y_k}^Tr_x)$ corresponds to $b(z|y_k)$ in Eq. (5).
>
> **The practical realization of e, b, c and d were not clear to me. The confusion could be resolved by presenting a pseudo-code of the practical implementation of the method.**
>
> Thank you. We have included the Tensorflow pseudo-code of C-SimCLR and C-BYOL below, which explicitly clarifies what the $e(.)$, $b(.)$, $c(.)$ and $d(.)$ distributions correspond to. We will add this to the appendix of the revision.
>
> ```python
> tfd = tensorflow_probability.distributions
>
> def contrastive_ceb(r_x,
>                     r_y,
>                     kappa_e=1024.0,
>                     kappa_b=10.0,
>                     beta=1.0):
>   """Contrastive version of CEB with von Mises-Fisher distributions.
>
>   Note that we can compute this loss in a “bidirectional” manner as
>   Bidirectional_loss = contrastive_ceb(r_x, r_y) + contrastive_ceb(r_y, r_x).
>   We use the same notation as the main paper.
>
>   Args:
>     r_x: Unit-length mean direction of view x. The expected shape is [B, dim].
>     r_y: Unit-length mean direction of view y. The expected shape is [B, dim].
>     kappa_e: A float. Concentration parameter of distribution e.
>     kappa_b: A float. Concentration parameter of distribution b.
>     beta: CEB beta for controlling compression strength (Equation 1).
>
>   Returns:
>     A tensor `loss`. The loss is per-sample.
>   """
>
>   batch_size = tf.shape(r_x)[0]
>   labels_idx = tf.range(batch_size)
>   # Labels are pseudo-labels which mark corresponding positives in a batch
>   labels = tf.one_hot(labels_idx, batch_size)
>   mi_upper_bound = tf.math.log(tf.cast(batch_size, tf.float32))
>
>   e_zx = tfd.VonMisesFisher(r_x, kappa_e)
>   b_zy = tfd.VonMisesFisher(r_y, kappa_b)
>   z = e_zx.sample()
>   log_e_zx = e_zx.log_prob(z)
>   log_b_zy = b_zy.log_prob(z)
>   i_xzy = log_e_zx - log_b_zy  # residual information I(X;Z|Y)
>   logits_ab = b_zy.log_prob(z[:, None, :])  # broadcast
>
>   # The following categorical corresponds to c(y|z) and d(x|z) in Equation 11:
>   cat_dist_ab = tfd.Categorical(logits=logits_ab)
>   h_yz = -cat_dist_ab.log_prob(labels_idx)
>   i_yz = mi_upper_bound - h_yz
>   loss = beta * i_xzy - i_yz
>   return loss
> ```
>
> ```python
>
> tfd = tensorflow_probability.distributions
>
> def byol_ceb_loss(omega,
>                   y,
>                   l_net,
>                   m_net,
>                   kappa_e,
>                   kappa_b,
>                   beta,
>                   byol_loss_weight):
>   """Compute loss for C-BYOL model.
>
>   The notation corresponds to Section 2.3 and Figure 2 of the paper.
>   This code presents an updated version of C-BYOL as described in the
>   general response.
>
>   Args:
>     omega: Unit-length mean direction of e(z|x). The expected shape is [B, dim].
>     y: Unit-length target projection output. The expected shape is [B, dim].
>     l_net: A transformation function. We choose a linear layer in this work.
>     m_net: A transformation function. We choose a two-layer MLP in this work.
>     kappa_e: A float. Concentration parameter of distribution e.
>     kappa_b: A float. Concentration parameter of distribution b.
>     beta: CEB beta for controlling compression strength (Equation 1).
>     byol_loss_weight: BYOL loss weight.
>
>   Returns:
>     A tensor `loss`. The loss is per-sample.
>   """
>   e_zx = tfd.VonMisesFisher(omega, kappa_e)
>   z = e_zx.sample()
>   s = l_net(z)
>   s = tf.math.l2_normalize(s, -1)
>
>   # The following regression corresponds to the unnormalized log probability d(y|z),
>   # as described in Section 2.3:
>   byol_loss = tf.reduce_sum(tf.math.square(s - y), axis=-1)
>
>   mu_prime = m_net(y)
>   mu_prime = tf.math.l2_normalize(mu_prime, -1)
>   b_zy = tfd.VonMisesFisher(mu_prime, kappa_b)
>
>   log_e_zx = e_zx.log_prob(z)
>   log_b_zy = b_zy.log_prob(z)
>   i_xzy = log_e_zx - log_b_zy
>
>   loss = byol_loss * loss_weight + beta * i_xzy
>
>   return loss
> ```
>
> **About the Lipschitz constant and robustness of a model, and relation to prior work.**
>
> Our experiments in Sec. 3.3 evaluate our models’ robustness to various domain shifts, including “natural adversarial examples” (ImageNet-A), synthetic corruptions (ImageNet-C), viewpoint (ObjectNet) and other natural variations (ImageNet-v2, ImageNet-R, ImageNet-Vid).
>
> This is related to Bruna and Mallat [7] who proposed a network that is stable to non-rigid deformations. Section 2.1 of [7] “formalizes the deformation stability condition as a Lipschitz continuity property”.
>
> Fazlyab et al [16] and Yang et al [48] propose methods for estimating the Lipschitz constants of neural networks, as “the Lipschitzness of $f$ is closely related to its robustness” [48]. However, these papers refer to robustness to adversarial examples, which is not the setting considered in this paper. We will therefore clarify this in the revision.

---

> > ### Comment · Reviewer_GtXm · 2021-08-26
> > **Concerns addressed**
> >
> > Thank you for taking the time to address my concerns.

---

### Author Response · Authors · 2021-08-10
**General Response**

We sincerely thank all our reviewers for the valuable feedback and their appreciation of  our empirical improvements over strong state-of-the-art baselines, thorough experiments and ablations, theoretical contributions, and clear presentation. We will improve the manuscript based on reviewers’ comments and address clarity concerns.

Two reviewers requested updated results with standard deviations. We include an updated version of Table 1 below, with mean and standard deviations computed over three runs. Note that post-submission, we have improved our C-BYOL model slightly, which was already achieving state-of-the-art results, by simplifying it: We chose a vMF distribution for $e(z|x)$ and $b(z|y)$, rather than the compound of vMF followed by a Gaussian as described in Lines 146-147 and 153-155. We will update Section 2.3 and Table 1 of the revised manuscript with these details.

|                            |          | Top 1 | Top 5 |
|----------------------------|----------|-------|-------|
| ResNet 50 (300 epochs)     | SimCLR   |  69.1±0.089     |  89.1±0.034     |
|                                             | C-SimCLR |   **70.1±0.177**    |  **89.6±0.099**     |
|                                             | BYOL     |   72.8±0.155    |  91.0±0.072     |
|                                             | C-BYOL   |  **73.6±0.039**     |   **91.5±0.080**    |
| ResNet 50 (1000 epochs)    | SimCLR   |  70.7±0.094     |  90.1±0.081     |
|                            | C-SimCLR |  **71.6±0.084**     |  **90.5±0.067**     |
|                            | BYOL     | 74.3±0.179      |  91.8±0.048     |
|                            | C-BYOL   |  **75.6±0.151**     |  **92.7±0.076**     |
| ResNet 50-2x (1000 epochs) | SimCLR   |  74.5±0.014     |  92.1±0.031     |
|                            | C-SimCLR |  **75.0±0.082**     |  **92.4±0.086**     |
|                            | BYOL     |  77.2±0.057    |  93.5±0.036     |
|                            | C-BYOL   |  **78.8±0.088**     |   **94.5±0.016**    |

---

### Decision · Program_Chairs · 2021-09-27

**Decision:**

Accept (Poster)

**Comment:**

This paper applies conditional entropy bottleneck (CEB) objective to two existing frameworks for self-supervised learning: SimCLR and BYOL.  The paper clearly demonstrates the advantages of this approach and is well-grounded in the existing theory on CEB. Adapting CEB to SimCLR and BYOL seems to be novel. Interestingly, presented approach can be applied in several similar settings, thus the results can be potentially impactful also beyond the main scope of the paper.